# Snake Venom Components as Therapeutic Drugs in Ischemic Heart Disease

**DOI:** 10.3390/biom13101539

**Published:** 2023-10-18

**Authors:** Erij Messadi

**Affiliations:** Plateforme de Physiologie et Physiopathologie Cardiovasculaires (P2C), Laboratoire des Biomolécules, Venins et Applications Théranostiques (LR20IPT01), Institut Pasteur de Tunis, Université Tunis El Manar, Tunis 1068, Tunisia; erij.messadi@pasteur.tn

**Keywords:** atherosclerosis, biologically active molecules, cardioprotection, hypertension, ischemic heart disease, myocardial infarction, snake venom

## Abstract

Ischemic heart disease (IHD), especially myocardial infarction (MI), is a leading cause of death worldwide. Although coronary reperfusion is the most straightforward treatment for limiting the MI size, it has nevertheless been shown to exacerbate ischemic myocardial injury. Therefore, identifying and developing therapeutic strategies to treat IHD is a major medical challenge. Snake venoms contain biologically active proteins and peptides that are of major interest for pharmacological applications in the cardiovascular system (CVS). This has led to their use for the development and design of new drugs, such as the first-in-class angiotensin-converting enzyme inhibitor captopril, developed from a peptide present in *Bothrops jararaca* snake venom. This review discusses the potential usefulness of snake venom toxins for developing effective treatments against IHD and related diseases such as hypertension and atherosclerosis. It describes their biological effects at the molecular scale, their mechanisms of action according to their different pharmacological properties, as well as their subsequent molecular pathways and therapeutic targets. The molecules reported here have either been approved for human medical use and are currently available on the drug market or are still in the clinical or preclinical developmental stages. The information summarized here may be useful in providing insights into the development of future snake venom-derived drugs.

## 1. Introduction

Ischemic heart disease (IHD) is the most common cause of death worldwide, accounting for 16% of total diseases [1,2]. IHD includes acute coronary syndrome (ACS), which is composed of unstable angina, myocardial infarction (MI) and sudden cardiac death [3]. MI, which is the major IHD manifestation and cause of death, induces myocardial necrosis and apoptosis in a short time, leading to organ dysfunction and HF with a poor prognosis [3]. IHD may arise for various causes, including hypertension, a major contributor to the high IHD prevalence, and atherosclerosis with coronary blood flow reduction to the heart muscle due to plaque formation and rupture [4,5]. Although myocardial reperfusion is essential to safeguard viable myocardium in MI patients, the process of restoring coronary blood flow to the ischemic myocardium can paradoxically induce myocardial injury in itself—a manifestation that has been termed “myocardial reperfusion injury” [6]. However, the progression of coronary revascularization techniques and angioplasty as well as the significant advances in the MI diagnosis and treatment, such as the development of novel antiplatelet and antithrombotic agents, remain insufficient [7]. This emphasizes the need to identify and develop new therapeutic strategies to reduce the prevalence of IHD as well as hypertension and atherosclerosis.

Some drug development strategies have shown promising results in the use of natural products, such as snake venom, to treat many diseases, including IHD [7,8,9,10]. The concept of using snake venom as a treatment for cardiovascular diseases (CVDs) emerged from the observation that snake bite envenoming is associated with a number of cardiovascular effects, including hypotension/hypertension, cardiotoxicity, MI, cardiac arrest, arrhythmias, coagulopathy and circulatory shock [11]. These potentially lethal manifestations caused by snake venom components have led to their use for thousands of years in traditional medicine as the basis of preparations and decoctions meant to cure cardiovascular disorders [12]. Since 1837, cobra venom has been used to treat vascular problems and ACS [12]. In the 20th century, the first animal toxin-derived drug, captopril, was designed based on bradykinin-potentiating peptides (BPPs) from the venom of the *Bothrops jararaca* snake and approved in 1981 for the treatment of hypertension [13]. Since then, other venom drugs have been developed based on or inspired by snake venom toxins, such as eptifibatide and tirofiban, and used to prevent heart attacks and thrombotic diseases [14]. Today, there is a renewed interest in snake-venom-based therapies, particularly due to the improvements in high-throughput approaches to the discovery of new toxin-based drugs. Some of these venom-based compounds are currently in clinical trials and many others may appear in the future as studies continue to investigate snake venom [15,16,17,18,19,20]. Snake venom is composed of highly selective and affine compounds [17,21] and mainly include proteins and peptides (90% to 95% of the venom dry weight) that target receptors (acetylcholine receptors, membrane transporters, enzymes) and ion channels (Na^+^, K^+^ and Ca^2+^ channels) [22]. In the CVS, snake toxins may have multiple targets such as cardiac muscle and vascular smooth muscle or the capillary vascular bed, and they include (i) proteins and peptides without enzymatic activity, such as BPPs, natriuretic peptides (NPs), disintegrins, C-type lectin-like proteins (CTLs), three-finger toxins (3FTx), vascular endothelial growth factors (svVEGFs), sarafotoxins (SFTXs), alternagin-C and cysteine-rich secretory proteins (CRISPs); and (ii) proteins with enzymatic activity, such as fibrinolytic enzymes (metalloproteinases and serine proteinases) and phospholipases A2 (PLA_2_s) [23,24] (Figure 1 and Table 1).

Overall, cardiovascular research on snake toxins has focused on three main areas related to the management of (i) MI and ACS, (ii) hypertension and (iii) atherosclerosis with the prevention of platelet aggregation and antithrombotic therapy. Therefore, this review describes the snake venom components that interact with the CVS and their use for the potential treatment of IHD and associated diseases (hypertension and atherosclerosis) by emphasizing their targets and mechanisms of action (Table 1). Venom toxins that do not directly interact with the CVS, such as proteins and peptides affecting blood cells and enzyme systems, are not discussed in this review. 

**Table 1 biomolecules-13-01539-t001:** Cardiovascular properties of snake venom components.

Proteins and Peptides	Molecular Mass (kDa)	Main Biological Target	Effects on Cardiovascular System		
			Effects in Ischemic Heart Disease	Effects on Blood Pressure	Effects on Atherosclerosis
Non-enzymatic toxins					
Bradykinin-potentiating peptides (BPPs)	1.5–3.0	Angiotensin-converting enzyme (ACE).	- Limiting myocardial injury and necrosis in MI and HF [9,25,26,27,28,29,30]. - Decreasing the risk of cardiac arrest (>20%) and increasing survival rates after MI [31]. - Improving endothelial vasomotor dysfunction and preventing myointimal proliferation after vascular injury [32,33,34,35].	- Lowering of BP through a decrease in the concentration of Ang II and an increase in the concentration of BK.	- Slowing the progression of arteriosclerotic lesions [36]. - Reducing aortic cholesterol content and the progression of carotid and coronary lesions due to a high-cholesterol diet [32,33,34,35].
Natriuretic peptides (NPs)	2.5–5.5	Natriuretic peptide receptor A, B and C.	- Cardioprotective effects in chronic HF [37,38]. - Improving cardiac function and reducing post-MI necrosis, fibrosis and inflammation through an NPR-A–cGMP-dependent pathway, downstream activation of mitochondrial K_ATP_ channels and inhibition of the mitochondrial permeability transition pore (mPTP) [15,17,18].	- Lowering of BP through a reduction in vascular resistance (due to a decrease in the influx of Ca^2+^ ions into muscle cells) and a decrease in the volume of circulating blood (due to an increase in the volume of excreted urine) - Endothelium-dependent vasorelaxation with increased NO production [39].	
Disintegrins	4–15	Platelet receptors αIIbβ3 integrin (glycoprotein IIb/IIIa receptors).	- Improving microvascular flow, reducing infarct size and the number of infiltrating platelets and post-IM leucocytes [40]. - Reducing the risk of IHD and preventing thrombotic complications and major adverse cardiac events in MI [41,42].		- Anti-atherogenic role by decreasing carotid atherosclerotic plaque inflammation and improving its stabilization in ACS patients [43,44]. - Improving the prognosis of ACS patients [45].
C-type lectins (CTLs)	30	Platelet receptors GPIb and GPIa/IIa (α2β1 integrin).			- Inhibiting platelet aggregation [46,47].
Three-finger toxins (3FTx)	5.2–8.0	Platelet receptors αIIbβ3 integrin, cholinergic receptors.	- Decreasing HR in vivo and ex vivo in isolated heart [48]. - Inducing a positive cardiac inotropic response [49]. - Suppressing the contractility and irreversible contracture of the myocardium, cardioprotection [49,50,51].	- Lowering BP and inducing hypotension. - Inducing vasodilation through a vasorelaxant effect on smooth muscles and preventing muscle contraction [52,53].	- Inhibiting platelet aggregation [54,55,56].
Vascular endothelial growth factors (svVEGFs)	24–25	Receptor tyrosine kinases VEGFR-1, VEGFR-2 and VEGFR-3.	- Decreasing infarct size through stimulation of VEGFR-2 receptors, ERK pathway activation and subsequent inhibition of mPTP opening and improvement of oxidative phosphorylation at the onset of reperfusion [16].	- Lowering BP through endothelium-dependent NO production [16,57].	
Sarafotoxins (SFTXs)	2.3–2.7	Endothelin type-A (ETA) and -B (ETB) receptors.	- Reducing infarct size and the incidence of arrhythmias [58,59,60,61] through the selective activation of ETB receptors prior to coronary occlusion, NO release and cardiomyocyte mitochondrial K_ATP_ channel activation [58,59,60,61].	- Inducing vasorelaxant and vasodilator effects [60,61].	
Alternagin-C	21.7	Integrin α2ß1 and VEGFR-2.	- Enhancing cardiac protection against hypoxia/reoxygenation-induced cardiomyocyte negative inotropism [62].		
Cysteine-rich secretory proteins (CRISPs)	23–25	Voltage-gated ion channels.	- Blocking smooth muscle contraction [63,64]. - Inhibiting high K-induced contraction [65]. - Inhibiting or activating aortic smooth muscle contraction [63,66]. - Blocking the ryanodine receptor [67].		
Enzymatic toxins					
Fibrinolytic enzymes (metalloproteinases and serine proteinases)	8–104	Fibrinogen (α- and β chains), fibrin.	- Reducing the mortality rate after IR, decreasing cardiomyocyte damage and cardiac biomarkers of ACS and improving myocardial function [68]. - Effective thrombolytic activity with a reasonable recanalization rate, low reocclusion rate and a low rate of bleeding complications [69].		- Stabilizing and inhibiting atherosclerotic lesions [70,71]. - Reducing total cholesterol content [71]. - Reducing the degree of stenosis of the coronary artery [71]. - Inhibiting platelet aggregation [72].
Phospholipases A_2_ (PLA_2_s)	13–14	Cell membrane, secretory PLA_2_ receptors.	- Cardiotoxicity [73,74]. - Bradycardia and atrioventricular block [75,76]. - Myocardial contracture [74,76,77].	- Lowering BP through the increase in plasma PGI_2_ and thromboxane A2 levels [78]. - Vasorelaxation [79,80].	- Anti-atherogenic activity [80,81].

The upper part of the table presents protein compounds without enzymatic activity: bradykinin-potentiating peptides (BPPs), natriuretic peptides (NPs), disintegrins, C-type lectin-like proteins (CTLs), three-finger toxins (3FTx), vascular endothelial growth factors (svVEGFs), sarafotoxins (SFTXs), alternagin-C and cysteine-rich secretory proteins (CRISPs). The lower part of the table presents protein compounds with enzymatic activity: fibrinolytic enzymes (metalloproteinases and serine proteinases) and phospholipases A_2_ (PLA_2_s). ACE, angiotensin-converting enzyme; ACS, acute coronary syndrome; Ang II, angiotensin II; BK, bradykinin; BP, blood pressure; MI, myocardial infarction; HF, heart failure; IHD, ischemic heart disease.

The review focuses on major toxins that directly affect the CVS, including four categories that are classified based on their cardiovascular therapeutic usefulness in humans: toxin-based approved drugs, toxin-based orphan drugs (i.e., synthetic pharmaceutical products that remain commercially undeveloped), toxins of interest in experimental and pre-clinical studies and toxins with limited experimental data (Table 2).

It should be noted that several molecules that were successful in animal assays were not approved in subsequent clinical trials. Several orphan drugs derived from snake venom also reached advanced clinical phases, but were stopped before market approval [82,83]. This is mainly due to the extrapolation of biological data across species, which is a key aspect of biomedical research and drug development. Caution should also be considered in interpreting animal experiments, as the use of in vitro assays in artificial conditions lacking a normal physiological environment and neurohumoral connections may influence study results and bias the subsequent in vivo data. This explains why some studies discussed in this review report conflicting results regarding venom compounds. 

**Table 2 biomolecules-13-01539-t002:** Classification of main snake venom toxins according to their usefulness in cardiovascular therapy or research.

Toxin-Based Approved Drugs				
Peptide/Protein	Origin	Drug	Mode of Action	Cardiovascular Indications
BPP-5a, BPP-9a (Bradykinin-potentiating peptides, BPPs)	*Bothrops jararaca*	Captopril/Enalapril	ACE inhibitor	Hypertension, MI, HF [84].
Barbourin (Disintegrin)	*Sistrurus m. barbouri*	Integrilin/Eptifibatide	GPIIb/IIIa antagonist	ACS and antithrombotic therapy [14,41].
Echistatin (Disintegrin)	*Echis carinatus*	Aggrastat/Tirofiban	GPIIb/IIIa antagonist	ACS and antithrombotic therapy [14,41].
Batroxobin (Fibrinolytic enzyme)	*Bothrops moojeni* *Bothrops atrox*	Defibrase	Cleavage of fibrinogen Aα subunit	Anticoagulant therapy in ACS [85,86].
Toxin-based orphan drugs				
Peptide/Protein	Origin	Drug	Mode of action	Cardiovascular indications
DNP (Natriuretic peptide)	*Dendroaspis angusticeps*	Cenderitide (CD-NP)	NPR-A and -B agonist	HF [37,38].
Fibrolase (Fibrinolytic enzyme)	*Agkistrodon contortrix contortrix*	Alfimeprase	Cleavage of fibrinogen α- and β-chains	Acute ischemic stroke, acute peripheral arterial occlusion, catheter occlusion [87,88,89].
Ancrod (Fibrinolytic enzyme)	*Calloselasma rhodostoma*	Viprinex	Cleavage of fibrinogen α-chain	Anticoagulant therapy in thrombosis [88,90].
Toxins of interest in experimental and pre-clinical studies				
Peptide/Protein family	Origin	Molecule	Mode of action	Cardiovascular indications
Natriuretic peptides	*Macrovipera lebetina*	Lebetin 2 (L2)	NPR-A and -B agonist	Increased BP, MI [15,17,18].
Disintegrins	*Rhinocerophis alternates*	Alternagin-C	GPIa/IIa (α2β1 integrin) antagonist VEGFR-2 inhibition	MI [62,91].
C-type Lectins (CTLs)	*Vipera palaestinae*	Vipegitide	GPIa/IIa (α2β1 integrin) antagonist	Antithrombotic therapy [14,92].
Three-finger toxins (3FTx)	*Dendroaspis jamesoni* *kaimosae*	Dendroaspin (mambin)	GPIIb/IIIa (Integrin α_IIb_β_3_) antagonist	Antithrombotic therapy [54].
Vascular endothelial growth factors (svVEGFs)	*Macrovipera lebetina*	ICPP	VEGF-A agonist	Increased BP, acute MI [16].
Sarafotoxins (SRTXs)	*Atractaspis engaddensis*	SRTX-6c	Endothelin receptors ETB agonist	MI [58,59,60,61].
Toxins with limited experimental data				
Peptide/Protein family	Origin	Molecule	Mode of action	Cardiovascular indications
Cysteine-rich secretory proteins (CRISPs)	*Gloydius blomhoffii*	Ablomin	L-type voltage-gated Ca^2+^ channel blockade	Hypertension [63,64].
Phospholipases A_2_ (PLA_2_s)	*Oxyuranus scutellatus* *Vipera palaestinae*	OSC3 PLA_2_ isoforms		Hypertension [93]. Atherosclerosis [81].

ACE, angiotensin-converting enzyme; ACS, acute coronary syndrome; DNP, *Dendroaspis* natriuretic peptide; HF, heart failure; ICPP, increasing capillary permeability protein; MI, myocardial infarction; NPR-A and -B, natriuretic peptide receptors A and B; VEGR-2, vascular endothelial growth factor receptor 2.

In the following part of the review, the order of toxins is presented as indicated in Table 1: proteins and peptides without enzymatic activity are described first (Section 2, Section 3, Section 4, Section 5, Section 6, Section 7, Section 8 and Section 9), followed by compounds with enzymatic activity (Section 10 and Section 11). 

## 2. Bradykinin-Potentiating Peptides

Bradykinin-potentiating peptides (BPPs) are proline-rich oligopeptides of 5 to 14 amino acid residues, predominantly generated by the enzymatic action of kallikreins on endogenous kininogen [94]. They are found in some snake venoms, mainly in *Bothrops* species, where they represent the first natural bradykinin (BK) agonists and ACE inhibitors [84,95]. In the organism, BPPs increase BK-induced hypotension and decrease vasopressor effects linked to angiotensin I (Ang I) by inhibiting ACE [96,97] (Figure 2). However, snake BPPs enhance BK-induced effects by interacting directly with BK receptors rather than inhibiting BK degradation through ACE inhibition [98]. Accordingly, BK receptor B2 (B2-R) stimulation induces vasodilation as well as anti-fibrosis, anti-inflammatory and anti-reactive oxygen species effects through various intracellular signaling pathways, while the inhibition of angiotensin II (Ang II) production prevents vasoconstriction, inflammation and cardiovascular damage [99,100]. The first BPP was isolated from the venom of the Brazilian viper *Bothrops jararaca* and then developed into a drug, captopril, against hypertension based on the structure of the peptides BPP-5a and BPP-9a [101]. Captopril was a significant milestone in several aspects: it was (i) the first drug developed from animal venom by converting the toxic effect into therapeutic action; (ii) one of the first examples of ligand-based drug discovery; (iii) the first ACE-targeting drug, becoming a blockbuster and inspiring many other ACE-inhibitor drugs based on the BPP-5a binding motif, like enalapril, lisinopril, quinapril, ramipril, trandolapril and moexipril [102,103]; and (iv) the drug that initiated the development of the next class of antihypertensive drugs, the Ang II-receptor antagonists (ARBs). Considering the therapeutic success of captopril, other snake BPPs have been investigated, particularly due to their specificity for the ACE C-domain. ACE has two homologous catalytic domains (N and C) with different substrate specificities: BK is hydrolyzed approximately equally by both catalytic domains, while Ang I is hydrolyzed approximately three times more efficiently by the C-domain than by the N-domain [104]. Thus, a C-domain-selective inhibitor would be more relevant as it mainly decreases the conversion of Ang I to Ang II by the C-domain and decreases the degradation of BK while preventing its accumulation by preserving ACE N-domain activity. As conventional ACE inhibitors, such as captopril, are not domain-selective and thus induce a higher risk of developing BK-mediated angioedema, this property of C-domain selectivity of BPPs makes them more beneficial than ACE inhibitors such as captopril [105]. Nevertheless, studies have shown that the complete inhibition of Ang I cleavage requires the blockade of both active sites of ACE [106,107,108]. Selectivity of action of the C-domain has been reported for certain snake BPPs, such as R-BPP and Y-BPP derived from *Azemiops feae* venom [109] and the decapeptide BPP-10c from *Bothrops jararaca* venom, which was found to be 400-fold more selective for the C-domain (Ki = 0.5 nM) than for the N-domain (Ki = 200 nM) [110]. In contrast, an opposite pattern was found for BPP-12b from the same snake, which was shown to be more selective for the N-domain (Ki = 5 nM) than for C-domain (Ki = 150 nM) [111]. Additionally, other BPPs with associated neuroendocrine functions have been discovered from the venom of *Bothrops*, *Crotalus* and *Lachesis* [112,113,114,115,116,117] and are suggested to belong to a novel class of endogenous neuropeptides [84,118]. 

### 2.1. BPPs and Cardioprotection in Ischemia–Reperfusion Injury

Initially indicated for the treatment of arterial hypertension, the use of BPP-based drugs such as captopril has been extended to prevent, treat or improve symptoms in conditions such as coronary artery disease and HF [29,30]. These drugs showed beneficial effects in limiting myocardial injury and necrosis in various models of coronary artery occlusion [9,25,26,27,28] and in clinical trials [29,30], where they decreased by more than 20% the risk of cardiac arrest and increased survival rates after MI [31]. Captopril has also been shown to slow the progression of arteriosclerotic lesions in clinical and preclinical studies by improving endothelial vasomotor dysfunction, preventing myointimal proliferation after vascular injury, reducing aortic cholesterol content and the progression of carotid and coronary lesions in high-cholesterol diet models [32,33,36,119]. Several lines of evidence indicate that the actions of ACE inhibitors in cardiac ischemia are due to their effect on the inhibition of endogenous BK degradation rather than their effect on Ang II inhibition [8,9,120,121]. BK has been suggested to be a key mediator in ischemic and pharmacological cardioprotective maneuvers via the inhibition of platelet aggregation and plasminogen activation, which could contribute to ACE-inhibitor-induced cardiovascular protection in IHD [9]. The BP effects of ACE inhibitors may also play a role; however, captopril has been found to be cardioprotective in hypertensive as well as in normotensive rats, suggesting that the cardiac effect of captopril may also involve the suppression of Ang II [119]. Recently, other BPPs, such as BPP-10c, which kinetically modulate argininosuccinate synthase activity with nitric oxide (NO) production in endothelial cells, have been shown to reduce hydrogen peroxide-induced oxidative stress [122,123]. Since post-MI reperfusion induces oxidative stress, NO deficiency and endothelial dysfunction leading to severe cardiac injury, BPP-10c could be further investigated as a promising therapeutic strategy for IHD. 

### 2.2. BPPs with Potential in Hypertension Therapy

A persistent increase in BP is a sign of hypertension that can lead, if untreated, to IHD [4,5]. To date, numerous antihypertensive drugs with different mechanisms of action have been developed. Snake venom BPPs decrease BP by enhancing the action of the endogenous vasodilator BK and inhibiting the vasoconstrictor Ang II [96,97] (Figure 2). However, it is not yet well established whether the cardiovascular effects of BPPs are only due to ACE inhibition [84]. Some studies have revealed that there are BPPs, such as BPP-10c, that possess non-ACE-inhibition mechanisms, such as the stimulation of argininosuccinate synthase, which leads to the production of NO in endothelial cells and a decrease in BP [84]. The BPP-5a peptide has also been shown to exert a long-lasting antihypertensive effect in spontaneously hypertensive rats (SHRs) via a unique target involving an NO-dependent mechanism [124,125,126]. In another study, the BPP-5a peptide was found to promote vasodilation by interacting with M1 muscarinic cholinergic receptors and the BK receptor B2-R and triggering NO synthesis by the endothelium [125]. Other members of the BPPs with a common hypotensive effect have been identified in different snake species, including *Bothrops jararaca* (BPP-7a) [127], *Bothrops insularis* [128], *Lachesis muta* (Lm-BPP 1–5) [115], *Agkistrodon bilineatus* [129], *Lachesis muta rhombeata* (LmrBPP-9) [130], and *Crotalus durissus cascavella* (BPP-Cdc) [131]. Different physiological mechanisms underlie the hypotensive effect of BPPs. Therefore, these studies contribute to the understanding of BP regulation and the identification of new therapeutic targets useful in the treatment of CVDs.

## 3. Natriuretic Peptides

Venom natriuretic peptides (NPs) are part of the NP family, which is composed of three mammalian NP isoforms, namely atrial NP (ANP), B-type NP (BNP), C-type NP (CNP) and an isoform originating from the *Dendroaspis* snake venom (DNP) [132]. These peptides are composed of approximately 20 to 50 amino acid residues and contain a conserved 17-residue disulfide ring [132]. NPs target membrane NP receptors (NPRs) of the reno-cardiovascular system to induce natriuresis, diuresis and vasorelaxation, thereby reducing blood pressure and volume [130,131] (Figure 2). NPs isolated from snake venom, although of a wide variety, nevertheless exhibit the same cardio-renal properties, allowing them to exert a broad spectrum of physiological effects that can be potentially used to treat CVDs [133]. Various snake venom NPs have been identified, such as Lebetin 2 (L2) from the blunt-nosed viper (*Macrovipera lebetina*) [134]; NP2_Casca from the cascavella rattlesnake (*Crotalus durissus cascavella*) [135]; PtNP-a from the eastern brown snake (*Pseudonaja textilis*) [136]; PaNP-c from the king brown snake (*Pseudechis australis*) [136]; PNP from the Persian horned viper (*Pseudocerastes persicus*) [137]; three natriuretic-like peptides, TNP-a, TNP-b, and TNP-c, from the venom of the inland taipan (*Oxyuranus microlepidotus*) [138]; KNP from *Bungarus flaviceps* [139] and Lm-CNP from *Lachesis muta* [115]. These venom peptides have been widely used as a promising basis for the design of cardiovascular drugs with improved activity, affinity and selectivity as well as longer stability and half-life [140,141].

### 3.1. Natriuretic Peptides and Cardioprotection in Ischemia–Reperfusion Injury

Recently, a drug developed by Novartis on the basis of its ability to potentiate endogenous NPs was commercialized to reduce the risk of cardiovascular death and hospitalization after HF [142]. Nesiritide is another drug used as a recombinant human BNP to improve the hemodynamic status of HF patients. However it induces severe adverse effects, such as hypotension, which currently limits its clinical use, highlighting the need for developing NP analogs without these side effects [143]. In this context, venom NPs have been used to develop analogues with the prospect of clinical application; among these, cenderitide is the most advanced [141]. It should be noted that NPs have the common property of inducing an NO increase and protein kinase G activation, which mediates their vasorelaxant effect [138,144]. Cenderitide (formerly CD-NP) was designed to overcome the hypotensive properties of NPs while improving cardiovascular function through the activation of the particulate guanylyl cyclases NPR-A and -B [141]. The chimeric peptide, composed of the 15-residue C-terminal tail of DNP and CNP, showed less adverse hypotensive effects than current NP drugs in preclinical studies as well as in the late phases of clinical trials [37,38], but the licensing agreement to further develop this peptide was recently terminated [82]. More recently, L2 from *Macrovipera lebetina* [134,145] has been proved to improve cardiac function and reduce necrosis, fibrosis and inflammation in a reperfused MI model [15,18]. Cardiac effects were shown to be mediated through an NPR-A–cyclic guanosine monophosphate (cGMP)-dependent pathway, downstream activation of mitochondrial K_ATP_ channels, and inhibition of the mitochondrial permeability transition pore (mPTP) at the time of reperfusion [15,17,18]. These results also showed that L2 can use a previously undiscovered mechanism involving M2 macrophage polarization and activation of the NPR-A–cGMP–Interleukin 10 axis to exert its post-MI anti-inflammatory effects [20]. Other ongoing experiments suggest that L2 could exert at least part of its preclinical effects via the involvement of the natriuretic receptor NPR-C, classically associated with the clearance of NPs, thus confirming the mechanistic role of NPR-C in NP cardiovascular effects [17,18]. Thus, these features demonstrate that venom compounds such as L2 may be of great therapeutic significance, especially in MIs with intense and prolonged inflammatory responses. These studies also provide novel insights into the mechanisms of myocardial repair triggered by NPs and could have good prospects for discovering new therapeutic targets and options in the treatment of IHD.

### 3.2. Natriuretic Peptides with Potential in Hypertension Therapy

Snake venom NPs exhibit the same hemodynamic properties as their mammalian counterparts. They decrease BP by inducing diuresis, natriuresis and vasodilation via endothelium-dependent vasorelaxation and by inhibiting the renin–angiotensin–aldosterone system [146]. Although NP hemodynamic properties have been proven, NPs are nevertheless not used clinically for the treatment of hypertension. In fact, potential NP-based drugs have been synthesized to overcome these hypotensive properties, as was the case with DNP [141]. However, several preclinical studies have demonstrated the vasorelaxant and vasodilator effect of natriuretic toxins. Fon instance, venom-derived L2 induces a dose-dependent decrease in the arterial BP in mice [18] (Figure 2). NP2_Casca was also shown to induce an endothelium-dependent relaxant effect on the thoracic aortic rings, likely involving K^+^ channels, along with a decrease in heart rate (HR) and arterial BP and an improvement in renal function [135]. Coa_NP, identified in *Crotalus oreganus abyssus* venom, also produces endothelium-dependent vasorelaxation in the thoracic aortic rings by increased NO production but without binding to the natriuretic receptor NPR-A [39]. For some other snake venom NPs, such as PaNP-c and PtNP-a, a dual mechanism was observed with ACE inhibition together with an increase in intracellular cGMP concentration [136].

## 4. Disintegrins

Snake venom disintegrins are small cysteine-rich proteins (4–15 kDa) that have been demonstrated to be potent inhibitors of various integrins. They can act on platelet integrin αIIbβ3, also known as glycoprotein IIb/IIIa receptors, to inhibit platelet aggregation in vitro and in vivo by blocking fibrinogen and von Willebrand factor binding to the GPIIb/GPIIIa complex on the surface of platelets [147] (Figure 3). Thus, antiplatelet drugs targeting this receptor could be indicated in the case of ACS and in percutaneous coronary interventions [14]. Snake venom disintegrins have RGD or KGD motifs that allow them to bind integrin and prevent fibrinogen binding to platelets [148,149]. An RGD tripeptide sequence is the main recognition site for the αIIbβ3 integrin receptor. As such, RGD disintegrins have been the most studied, and research on these have resulted in the design and synthesis of novel antiplatelet pharmaceutical compounds [148]. Typical antiplatelet drugs found in clinics to treat ACS include Integrilin (eptifibatide), a heptapeptide derived from barbourin, a protein found in the venom of the American Southeastern pygmy rattlesnake (*Sistrurus miliarius barbourin*) and Aggrastat (tirofiban), a small molecule based on the structure of echistatin that is found in the venom of the saw-scaled viper (*Echis carinatus*) [14]. Several other disintegrins with common antiplatelet aggregation potential have been purified from venoms of snakes such as the common bamboo viper (*Trimeresurus gramineus*) [147], the Malayan pit viper (*Calloselasma rhodostoma*) [150], the Halys pit viper (Gloydius halys) [151] and the fer-de-lance (*Bothrops asper*) [152]. These disintegrins have been described as highly potent and selective GPIIb/IIIa antagonists in vitro and in vivo [147,153]. In addition to their antiplatelet effect, some disintegrins have angiogenic properties. For instance, CC5 and CC8, two highly homologous disintegrins isolated from the North African viper *Cerastes cerastes* venom, inhibit angiogenesis by disrupting αvβ3 and α5β1 binding in vitro in human microvascular endothelial cells (HMEC-1) and human brain microvascular endothelial cells (HBMEC) [154], ex vivo in a rat aortic ring assay and in vivo in a chick embryo chorioallantoic membrane model (CAM). These effects, which require an RGD-loop disintegrin, have been shown to be mediated by pro-apoptotic pathways involving the downregulation of the FAK/AKT/PI3K axis and caspase activation. Similarly, alternagin-C, a disintegrin-like peptide isolated from *Bothrops alternatus* snake venom, impairs angiogenesis following binding to α2β1 integrin and inhibition of VEGF/VEGFR-2 signaling [91].

### 4.1. Disintegrins and Cardioprotection in Ischemia–Reperfusion Injury

The snake venom disintegrins-based drugs Integrilin (or eptifibatide) and Aggrastat (or tirofiban) designed from barbourin and echistatin, respectively, are the most frequently studied GPIIb/IIIa inhibitors [41]. They were approved by the FDA in 1988 for the treatment of ACS [14]. Both drugs are specific αIIbβ3 antagonists, designed based on the KGD pharmacophore for eptifibatide and the RGD pharmacophore for tirofiban. They exert their actions by preventing the binding of fibrinogen to αIIbβ3 of human platelets and thereby inhibiting platelet aggregation and thrombus formation (Figure 3). In preclinical studies, eptifibatide and tirofiban have been shown to improve microvascular flow, reduce the infarct size and decrease the number of infiltrating platelets and leucocyte recruitment and accumulation in the ischemic myocardium at the onset of reperfusion [40]. Clinically, these drugs reduce the risk of IHD and prevent thrombotic complications and major adverse cardiac events that can occur during and after percutaneous coronary intervention [41,42]. However, despite the beneficial effects of GPIIb/IIIa inhibitors, they are not included in the current therapeutic protocol for MI patients. Due to the risk of bleeding complications associated with GPIIb/IIIa inhibitors, only patients with a high risk of thrombosis are currently treated with GPIIb/IIIa inhibitors [40]. Consequently, numerous preclinical studies have focused on the therapeutic potential of other snake-venom disintegrins in IR injury. Some of them are under experimental study and have good prospects for the treatment of IHD and related diseases, such as alternagin-C from the *Bothrops alternatus* venom, which has been found to protect cardiomyocytes against hypoxia–reoxygenation-induced negative inotropism [62].

### 4.2. Disintegrins in Atherosclerosis Therapy

Most cases of acute MI are caused by the rupture of an atherosclerotic plaque associated with subsequent thrombus formation [155]. Atherosclerosis is a fibroproliferative and inflammatory process occurring in arterial walls in response to several factors, such as hypertension and hypercholesterolemia. Low-density-lipoprotein (LDL) cholesterol is an important causal risk factor for atherosclerotic CVDs. In reperfused MI, elevated LDL cholesterol promotes thromboinflammation through excess microvascular endothelial von Willebrand factor and platelet adhesion, resulting in less microvascular reflow and a larger infarct size [156]. The rupture of an atherosclerotic plaque leads to platelet aggregation, thrombosis and an increased risk of ACS [155,157]. Thus, in the presence of elevated LDL cholesterol, therapies that suppress endothelial-associated von Willebrand factor and platelet aggregation may promote recovery of the left ventricular function and protect against IR and subsequent remodeling. In this context, chronic treatment with tirofiban and other antiplatelet drugs has been shown to decrease carotid atherosclerotic plaque inflammation and reduce serum levels of inflammatory markers such as Hs-CRP, IL-6 and sICAM-1 in ACS patients [43,44]. In addition, the combination of tirofiban and other drugs has been demonstrated to have a synergistic effect and significantly attenuate the inflammation response and improve the prognosis of ACS patients [45], which may lead to stabilization of the plaque, highlighting an anti-atherogenic role of disintegrin-based antiplatelet drugs.

## 5. C-Type Lectin-like Proteins

C-type lectins (CTLs) are Ca^2+^-dependent carbohydrate-binding proteins that share structural homology in their carbohydrate-recognition domains (CRDs). CTLs are widely represented in nature and are classified into several types according to their structure, cellular target and mode of action [47]. However, those found in snake venom mainly present a heterodimeric structure, with two subunits α and β that are responsible for the inhibitory effect on platelet function [47]. Heterodimeric CTLs have been shown to inhibit platelet aggregation by inhibiting receptor GPIb binding to von Willebrand factor platelet [47] (Figure 2). Among the toxins interacting with the GPIb receptor, several molecules have been characterized, such as lebecetin from *Macrovipera lebetina* [158], CHH-A and B from *Crotalus horridus horridus* [159], echicetin from *Echis carinatus* [160], TSV-GPIB-BP from *Trimeresurus stejnegeri* [161], tokaracetin from *Trimeresurus tokarensis* [162] and agkistin from *Agkistrodon acutus* [163]. Other snake CTLs inhibit platelets by binding to GPIa/IIa platelet receptors (α_2_β_1_ integrin), such as EMS16 from *Echis multisquamatus* [164] and rhodocetin from *Calloselasma rhodostoma* [165]. Recently, vipegitide, developed from CTL derived from *Vipera palaestinae* venom and targeting the GPIa/IIa receptor, was shown to be effective in inhibiting platelet adhesion to collagen, but this work is still in the preclinical stage [14,92]. Overall, despite the supportive role that CTLs play in inhibiting platelet adhesion, there is relatively poor drug development targeting snake-venom CTLs with respect to arterial thrombosis [14]. Indeed, due to their interaction with various integrins, CTLs have been mainly studied in angiogenic pathologies such as cancer and ocular neovascularization [19,166,167]. This is the case for lebecetin which has, in addition to its antiplatelet properties [46], anti-integrin (α_v_β_3_, α_v_β_5_ and α_5_β_1_) effects [166], and which is currently patented for the treatment of neovascular disorders such as age-related macular degeneration (AMD) and diabetic retinopathy [19].

## 6. Three-Finger Toxins

Three-finger toxins (3FTx) represent one of the most abundant families of snake-venom toxins. They contain 57–82 amino acid residues, characterized by three β-structural loops extending from a compact hydrophobic core that is stabilized by 4–5 disulfide bridges [168]. The pharmacological effects of 3FTx are extremely varied due to the wide range of molecular targets they recognize, such as L-type calcium channels and nicotinic and muscarinic acetylcholine receptors, which leads to different biological properties, particularly in the CVS [51,169]. It has been found that 3FTx may induce cardiac arrest [170] and changes in BP [52,171] and HR [48], inhibit platelet aggregation [172], blood coagulation [173,174], or cell adhesion [175]. Currently there are increasingly new sequences of 3FTx being available in public databases, which should provide new opportunities for the development of therapeutics or research probes targeting the CVS [51].

### 6.1. Three-Finger Toxins and Cardioprotection in Ischemia–Reperfusion Injury

Several 3FTx may be of interest in reducing IR injury following MI [51]. Some of them exert antiplatelet effects, such as the compound KT-6.9, purified from the monocled cobra (*Naja kaouthia*) and acting via the ADP receptors located at the surface of the platelets [176]. Other 3FTx include muscarinic toxins (MTα from *Dendroaspis polylepis* [177]), β-adrenergic toxins (β-cardiotoxin from the king cobra (*Ophiophagus hannah*) [48]), cholinergic toxins [178] or acid-sensing ion channel (ASIC)-inhibiting toxins (mambalgins from mambas (*Dendroaspis*) [179]) that may be potentially useful in the treatment of chronic HF and BP disorders, although there is no strong evidence of their applicability in IHD. Certain 3FTx from *Naja naja siamensis* with high toxicity for heart and vessels, have been found to induce a positive inotropic response [49]. This group of toxins could be potentially useful in cardiac conditions with decreased pumping function of the heart, particularly because there is only one pharmacological class of cardiotonic drugs, i.e., cardiac glycosides, used in the chronic HF with reduced left ventricular ejection function [180].

### 6.2. Three-Finger Toxins in Hypertension Therapy

Some 3FTx family members such as calciseptine [52], FS2 [171], C10S2C2 [181] and S4C8 toxins [182], all purified from mamba snakes, have the property of acting on various ion channels such as the L-type Ca^2+^-channel blockade, which leads to a vasorelaxant effect on smooth muscles, causing vasodilation and a reduction in BP [52,53] (Figure 2). In humans, L-type Ca^2+^-channel blockers are used for the treatment of hypertension, angina and arrhythmia [183]. However, they lack selectivity and are associated with serious adverse events. In this context, synthetic analogues based on calciseptine and FS2 have been generated via the identification of the amino acid residues of the toxin acting as a potential attachment region at the Ca^2+^-channel-receptor site. These analogues have been proven to have conformational and molecular properties similar to those of nifedipine, a 1,4-dihydropyridine L-type Ca^2+^-channel blocker currently used as antihypertensive treatment [184].

### 6.3. Three-Finger Toxins in Atherosclerosis Therapy

Platelets, which are major players in the atherosclerosis process, express receptors that are molecular targets of many snake toxins, including 3FTx. Several 3FTx that inhibit platelet aggregation, such as dendroaspin (also named mambin) [54] and S5C1 [55] from *Dendroaspis jamesoni kaimosae*, thrombostatin from *Dendroaspis angusticeps* [56] and γ-bungarotoxin from the venom of *Bungarus multicinctus*, have an RGD tripeptide sequence in their structure, which is crucial for binding to platelet integrin α_IIb_β_3_ and blocking platelet aggregation from binding to fibrinogen at a nanomolar level (Figure 3). Changes in the optimal conformation of the RGD tripeptide have been shown to modify the antiplatelet activity of 3FTx-related proteins [185]. Recently, other 3FTx, such as TA-bm16 and NTL2 identified from *Bungarus multicinctus* [51], have also been shown to potentially display a similar function.

## 7. Vascular Endothelial Growth Factor

VEGF-like proteins derived from snake venom, svVEGFs or VEGF-F, are a subgroup of the VEGF family that includes VEGF-A, VEGF-B, VEGF-C, VEGF-D, VEGF-E (viral VEGF) and placental growth factor (PlGF) [24]. VEGF binds to endothelial tyrosine kinase cell receptors known as VEGFR-1 and -2 to mediate its angiogenic, mitogenic and anti-apoptotic activities [186]. Receptor-binding assays demonstrated that svVEGFs are highly specific ligands [57], selectively interacting with VEGFR-2, as is the case with vammin from the venom of the sand viper *Vipera ammodytes*; or with VEGFR-1, as for Tf-svVEGF and Pm-VEGF from the venoms of *Trimeresurus flavoviridis* and *Protobothrops mucrosquamatus*, respectively [187]. After an acute MI, VEGF induces angiogenesis by initiating the reactive oxygen species–endoplasmic reticulum stress–autophagy axis in the vascular endothelial cells [188]. Therapeutic angiogenesis has shown promising results in preclinical studies in acute MI models by inducing safe, effective and sustained myocardial angiogenesis, increasing perfusion of the infarct border zone and the ventricular ejection fraction [189]. Since after MI, endogenous angiogenesis cannot maintain normal capillary density [190]. Thus, VEGF-based treatment strategies may help restore adequate vasculogenesis and be useful as a pro-angiogenic therapy in IHD. In mice IR experiments, svVEGF purified from *Macrovipera lebetina* venom and called Increasing Capillary Permeability Protein (ICPP), was found to exert cardioprotective effects by reducing the infarct size through the activation of VEGFR-2 receptors and the ERK pathway, and the subsequent inhibition of mPTP opening at the onset of reperfusion [16]. This protein and others, such as vammin and VR-1 [57], also induce endothelium-dependent vasorelaxation via the release of NO and PGI_2_ [191] thus leading to hypotension, higher than that induced by natural VEGF (Figure 2) [16,57]. Other VEGF-like proteins such as heparin-binding dimeric hypotensive factor (HF) purified from Aspic viper (*Vipera aspis aspis*) venom have also been demonstrated to induce hypotensive effects [192].

## 8. Sarafotoxins

Sarafotoxins (SRTXs) are a family of small 2.5 kDa peptides that are overproduced in *Atractaspis* venoms. They have a common structural pattern consisting of 21 amino acids and two invariant disulfide bridges between cysteines 1 and 15 and cysteines 3 and 11. A longer isoform, SRTX-m of 24 amino acids with three additional C-terminal residues, has, however, been isolated from the venom of *Atractaspis microlepidota microlepidota* and shows high sequence homology to SRTX-b [193]. SRTXs are potent vasoconstrictor peptides with approximately 60% sequence homology and functional identity to the mammalian vasoconstrictor hormones endothelins [194,195]. Several isoforms of SRTXs can coexist within the same venom with different effects, such as SRTX-a, -b, and -c that are all purified from the snake venom *Atractaspis engaddensis* and that present different vasoactive effects linked to their primary structure [196]. SRTXs also exhibit opposite cardioprotective effects. For example, unlike other SRTX isoforms, SRTX-6c, a specific agonist of the ETB endothelin receptor, exerts vasorelaxant and vasodilatory effects [60,61]. Administration of SRTX-6c prior to coronary occlusion has been shown to significantly reduce infarct size and the incidence of arrhythmias in several experimental models [58,59,60,61] through the selective activation of ETB receptors, NO release and subsequent cardiomyocyte mitochondrial K_ATP_-channel opening during IR [59]. Conversely, SRTX-b, whose effects are mainly associated with extracellular calcium input via L-type Ca^2+^ channels, causes cardiac arrest and death in mice within minutes of intravenous administration [197]. Despite their overall detrimental effects on the CVS related to their vasoconstrictor and arrhythmogenic properties, SRTXs are used in basic research for endothelin receptor labeling and in the development of vasospasm models [198].

## 9. Cysteine-Rich Secretory Proteins

Snake venom cysteine-rich secretory proteins (CRISPs) are a family of 23–25 kDa proteins containing eight disulfide bonds and are characterized by a single polypeptide protein with a molecular weight between 20 and 30 kDa [199]. CRISPs are present in the venoms of viperids, elapids and colubrids and absent in the venoms of atractaspidids and those of certain elapids such as coral snakes [22]. The first CRISP group to have been characterized includes ablomin from *Gloydius blomhoffii*, triflin from *Protobothrops flavoviridis*, latisemin from *Laticauda semifasciata* and tigrin from *Rhabdophis tigrinus*, which have the non-enzymatic inhibitory activity of various membrane channels involved in the regulation of vascular tone, such as voltage-gated L-type Ca^2+^ channels and high-conductance calcium-activated potassium (BKCa) [63,64]. Other members of the CRISP family, such as natrin, block the skeletal isoform of the ryanodine receptor [67] and KV1.3 voltage-gated potassium channels [65]. Other CRISP toxins have been shown to act on other cellular targets, such as inhibiting angiogenesis [200,201] and increasing blood vascular permeability in vivo and in vitro [202]; however in most cases, these proteins have not been experimentally isolated or characterized, and their targets and biological roles remain poorly understood.

## 10. Fibrinolytic Enzymes

Apart from antiplatelet compounds that have antithrombotic activity, snake venom also includes important thrombolysis molecules such as fibrinolytic enzymes involved in fibrinolysis, which is the process of dissolving blood clots (Figure 3). These enzymes break down fibrin-rich clots and help prevent additional clot formation by their action on fibrinogen (Figure 3). Given that these enzymes should not be inactivated by mammalian blood inhibitors, a significant number of studies have focused on their identification and characterization from snake venom, particularly for the development of thrombolytic enzymes to treat occlusive thrombi [203]. Two classes of venom fibrinolytic enzymes have been identified, the metalloproteinases and serine proteinases, with the same biological effects but different mechanisms of action by targeting different amino acid sequences of fibrinogen. Several fibrinolytic enzymes have been isolated and purified from snake venom and show potential use in thrombolytic therapy, including fibrolase, a fibrinolytic metalloproteinase isolated from *Agkistrodon contortrix* venom. Fibrolase possesses three disulfide bonds, is non-glycosylated and binds an intrinsic zinc atom that is essential for activity and structural integrity. It acts directly on fibrinogen/fibrin in both arteries and veins by cleaving the α- and β-chains of fibrinogen but not the γ-chain [87,88,89]. The recombinant fibrolase alfimeprase has been shown to have the same in vivo and in vitro activities as native fibrolase and to be much more potent than plasminogen activators [83]. Hence, treatment with alfimeprase would be more beneficial, as systemic plasminogen activation can lead to serious side effects, such as intra-cranial hemorrhaging [204]. Since then, other fibrinolytic enzymes have been identified and purified from other species such as lebetase, a serine beta-fibrinolytic proteinase from *Macrovipera lebetina* [205], basiliscus fibrases from the venom of the Mexican West Coast rattlesnake (*Crotalus basiliscus*) [72] and graminelysin I from *Trimeresurus gramineus* [205].

### 10.1. Fibrinolytic Enzymes and Cardioprotection in Ischemia–Reperfusion Injury

Following MI, thrombolytic therapy is the primary approach used for reperfusion to restore blood flow in the thrombus-obstructed coronary artery. The fibrinolytic enzyme-based drug for the treatment of thrombotic diseases is Defibrase, which has thrombin-like activity and is developed from batroxobin, derived from the venom of the Brazilian lancehead snake (*Bothrops moojeni*) [85]. It exhibits anticoagulant properties by converting fibrinogen into fibrin through the cleavage of the α-chain [86]. In clinical trials in MI patients, Defibrase was effective in inducing significant clot thrombolysis and coronary artery recanalization, with a low risk of recurrence and bleeding complications [69]. In preclinical studies, batroxobin was shown to reduce the mortality rate after IR, improve cardiac contractility and decrease myocardial injury and levels of ACS biomarkers [68]. Since Defibrase, other snake defibrinogenating agents have been identified. Reptilase is another trade name for a thrombin-like serine protease that was similarly developed from the common lancehead (*Bothrops atrox*), tested in animals, and used in patients to achieve coronary reperfusion [206]. In contrast, fibrolase (Alfimeprase) and ancrod (Viprinex) reached phase III clinical trials but failed to have marketing approval, since they did not meet the expected therapeutic endpoints, which led to the termination of their development [90]. However, experimental studies carried out previously on these molecules have shown the beneficial effects of fibrolase in the canine model of carotid artery thrombosis [88]. In this model, the occlusive thrombus was lysed within 6 min of initiating fibrolase infusion. Treatment with ancrod, however, had no effect on infarct size in the rabbit coronary ligation model [207] and did not protect the myocardium from reperfusion injury after acute MI in dogs [208].

### 10.2. Fibrinolytic Enzymes in Atherosclerosis Therapy

Snake-venom-derived fibrinolytic enzymes such as batroxobin and the batroxobin based-drug Defibrase may have an effect on atherosclerotic lesions. The first was found to possess the action of stabilizing the atherosclerotic plaque in a rabbit model of atherosclerosis [70]. From these results, the atherosclerotic plaque in the batroxobin-treated groups tended to be static four weeks after treatment. For the batroxobin-based drug Defibrase, experimental data have shown that treatment with Defibrase can significantly inhibit atherosclerosis in rabbits [71]. In this study, Defibrase decreased the mean aortic plaque area in the thoracic and abdominal aorta as well as the total cholesterol content compared with control animals. In addition, Defibrase treatment reduced the number of plaques at the small artery openings in the thoracic aorta and the abdominal aorta. Cross sections performed at the upper third of all hearts showed that Defibrase administration reduced the degree of stenosis of coronary artery branches in the heart wall compared with untreated animals.

## 11. Phospholipases A_2_

PLA_2_s are the most abundant proteins found in Viperidae snake venom and one of the major toxic components with a broad spectrum of pharmacological effects [209]. There are at least 15 distinct PLA_2_s groups, clustered into four major enzyme types: cytosolic PLA_2_s (cPLA_2_), Ca^2+^-independent PLA_2_s (iPLA_2_), secreted PLA_2_s (sPLA_2_) and platelet-activating factor acetylhydrolases (PAF-AH) [210]. Snake venom phospholipases belong to the sPLA_2_ family and exhibit diverse effects, including neurotoxicity, cardiotoxicity, myotoxicity, inhibition of blood coagulation, modulation of platelet function, as well as antiangiogenic activity independent of their enzymatic function [211]. In the CVS, snake venom PLA_2_ can cause myocardial structural and functional alterations [73,74,75,76]. This cardiotoxic activity can differ considerably between PLA_2_s of different venoms, e.g., PLA_2_s from *Ophiophagus hannah* and *Naja nigricollis* cause myocardial damage [74,76,77], in contrast to the PLA_2_ from the venom of *Naja atra,* which lacks cardiotoxicity [77]. PLA_2_s from certain snake species, such as the Elapinae and Viperinae subfamilies can also decrease BP through the hydrolysis of membrane glycerophospholipids and the subsequent release of arachidonic acid [212], a precursor of cyclooxygenase metabolites (prostaglandins or prostacyclin (PGI_2_)) (Figure 2). PLA_2_ fractions from *Vipera russelli* venom injected intravenously into animals induced marked vasodilation and a decrease in the mean arterial pressure and renal vascular resistance, partly due to increased plasma PGI_2_ and thromboxane A2 levels and a decrease in plasma renin activity [78,79]. The hypotensive and vascular relaxing effects of PLA_2_s have also been reported with toxins from the Australasian elapid Papuan taipan (*Oxyuranus scutellatus*) venom, such as the OSC3 peptide, whose antihypertensive activity is mediated by cyclooxygenase metabolites, BK and H1-receptors [93]. Some reports also suggested anti-atherogenic activity of snake PLA_2_s such as crotoxin from the South American rattlesnake (*Crotalus durissus terrificus*) [80]. In human umbilical vein endothelial cells (HUVEC), crotoxin has been found to downregulate the increased levels of adhesion molecules, inflammatory cytokines and oxidative stress, which are characteristic features of the early stages of atherosclerosis [80]. Another work also reported that *Vipera palaestinae* venom could potentially have anti-atherosclerotic action, as it rapidly decreases serum cholesterol levels in humans, with changes in lipoprotein transport and metabolism likely caused by the PLA_2_ component of this snake [81] (Figure 3).

Overall, due to their cytotoxic and inflammatory potential [79], pharmacological applications of snake venom PLA_2_s remain limited and are instead used in basic research to help elucidate mechanisms of action of PLA_2_s. However, the growing interest in the design of therapeutic drugs based on low-mass peptides has encouraged the design of small synthetic peptides from PLA_2_s. In particular, the identification of regions of PLA_2_, mainly in the C-terminal domain, with toxic or therapeutic functions, has contributed to the development of PLA_2_ analogues that may have high therapeutic potential [213]. Other work has also demonstrated that small subunits of venom PLA_2_s, such as Hemilipin2, derived from the *Hemiscorpius lepturus* scorpion, can mediate the biological effects of the entire protein without inducing cytotoxic effects [214]. These findings could serve as a starting point for the design of a new generation of low-molecular-mass PLA_2_s for the treatment of IHD without the adverse effects associated with high-molecular-mass native molecules.

## 12. Discussion

Despite the significant decrease in coronary mortality over the past 20 years, IHD is still the leading cause of death, reaching 9 million deaths worldwide [1,2,215]. This mortality decrease is mainly linked to considerable progress in medical therapies, far ahead of the effects of revascularization [215]; this, in particular, is thanks to the use of four major therapeutic classes with clinically proven benefits: platelet aggregation inhibitors, beta blockers, ACE inhibitors and statins. To further reduce the high mortality rate induced by IHD, new therapeutic alternatives are still needed, particularly through the exploration of snake venom, which is one of the main natural sources of therapeutic peptides and proteins. The advent of new technologies in the field of biomedical research, such as “omics” (genomics, transcriptomics, proteomics, metabolomics) and bioinformatics [17], has demonstrated promising new prospects for the use of snake venom toxins as a basis for drug discovery. This has led to the marketing of several approved drugs designed or inspired by the chemical structure of snake venom bioactive molecules, as well as a plethora of other toxins of pharmacological, biomedical and biotechnological interest (Table 2). However, despite these promising venomic advances, there are still challenges as well as limitations in the development of therapeutic drugs from snake venom toxins, due to the significant gap between the increased number of venom compounds with relevant pharmacological properties and the few compounds approved and used in human therapeutics. The main limitation lies in the development of drugs from high-molecular-weight toxins generally being considered more immunogenic than their low-molecular-weight counterparts and more difficult to produce and synthesize [216]. In this respect, in recent years, research has increasingly focused on peptide-based drugs as therapeutics for the management of several diseases.

Technological advances in venom toxin purification and characterization as well as new chemical synthesis methodologies have opened up new opportunities in the discovery of low-mass venom components that could not previously be detected or identified, and in the development of peptidomimetics for therapeutic purposes [217]. Small toxins can be easily produced and engineered to develop peptidomimetics through the introduction of structural modifications that may improve their therapeutic potency or confer resistance to proteolytic degradation, key factors that need to be optimized prior to clinical trials [218]. In turn, the use of synthetic peptides has challenged the design and development of small peptidomimetics from high-molecular-weight proteins [213]. Considering that large molecules represent the majority of the snake venom protein content (i.e., 90 to 95%) [219], it is therefore worth developing low-mass peptides to overcome the deleterious effects of native proteins while preserving their beneficial pharmacological properties [213].

## 13. Conclusions

Despite the complex challenges in the field of toxinology, snake-venom-derived molecules remain a highly promising source of lead compounds and novel drugs due to their high selectivity toward cellular targets, stability and efficacy compared with their human counterparts. Since the successful discovery in 1981 of the snake venom drug, captopril, several examples from venomic research, as reported in this review, have proven the benefit of using venom molecules rather than their human counterparts. In certain cases, venom molecules are developed to counter unexpected adverse effects found with human components such as the venom DNP-based cenderitide developed to overcome the deleterious hypotensive effects of nesiritide, the recombinant human BNP used as a drug for the treatment of HF. Another example is that of L2, a snake venom BNP-like peptide that has been shown to be more potent than the human counterpart, with additional effects and a novel mechanism of action not observed with human NPs [15,17,18,20]. The venom-derived CTL, lebecetin, approved in preclinical studies, has been patented for its high selectivity and potent ability, compared with other synthetic family members, to simultaneously induce multiple cellular targets for the treatment of neovascular eye diseases [19]. Other toxins have the ability to escape human enzyme systems, thus being useful as diagnostic tools, e.g., for the assay of coagulation factors and for the study of hemostasis [220].

## Figures and Tables

**Figure 1 biomolecules-13-01539-f001:**
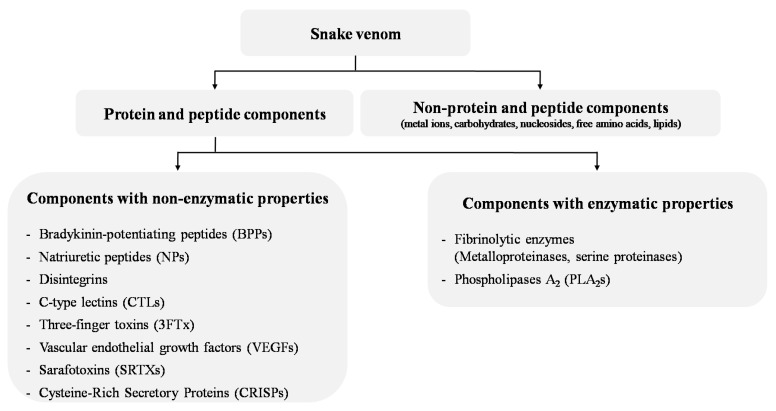
Main snake venom components with cardiovascular activities.

**Figure 2 biomolecules-13-01539-f002:**
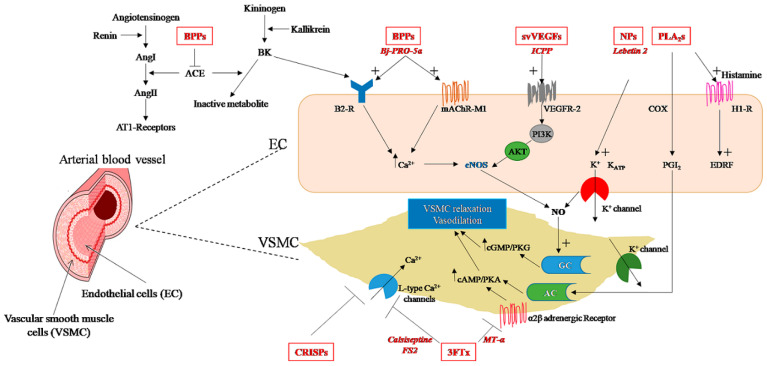
Snake venom toxins with hypotensive properties (highlighted in red): mechanisms of action in endothelial cells (EC) and vascular smooth muscle cells (VSMC) (details are included in the text). AC, adenylyl cyclase; ACE, angiotensin-converting enzyme; AKT, protein kinase B; AMP, adenosine monophosphate; Ang, angiotensin I; Ang II, angiotensin II; AT1-receptors, Angiotensin II receptor type 1; B2-R, bradykinin receptor B2; BK, bradykinin; BPPs, bradykinin-potentiating peptides; cAMP, cyclic adenosine monophosphate; cGMP, cyclic guanosine monophosphate; COX, cyclooxygenase; CRISPs, cysteine-rich secretory proteins; EDRF, endothelium-derived relaxing factor; eNOS, endothelial nitric oxide synthase; GC, guanylate cyclase; H1-R, histamine H1 receptor; mAChR-M1, M1 muscarinic acetylcholine receptor; MT-α, muscarinic toxin; NO, nitric oxide; PGI_2_ prostaglandin I2 (prostacyclin); PI3K, phosphoinositide 3-kinase; PKA, protein kinase A; PKG, protein kinase G; PLA_2_s, phospholipases A_2_; NPs, snake venom natriuretic peptides; PI3K, phosphatidylinositol-3-kinase; svVEGFs, snake venom vascular endothelial growth factors; 3FTx, three finger toxins; VEGFR-2, vascular endothelial growth factor receptor type 2.

**Figure 3 biomolecules-13-01539-f003:**
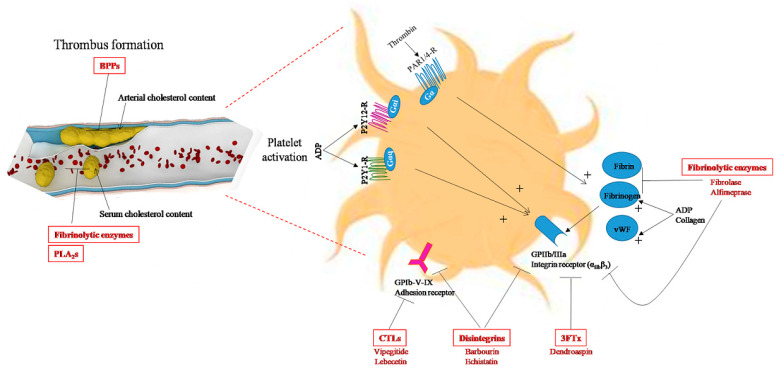
Snake-venom toxins with anti-atherogenic properties (highlighted in red). Snake-venom-derived molecules may exert anti-atherogenic effects by lowering cholesterol levels in the blood or in the arterial wall (BPPs, fibrinolytic enzymes, PLA_2_s), or by inhibiting arterial thrombosis by blocking platelet aggregation (disintegrins, CTLs, 3FTx, fibrinolytic enzymes,) or by activating fibrinolysis in blood clots (fibrinolytic enzymes: metalloproteases and serine proteases). Members of the disintegrin and fibrinolytic enzyme families are currently the main toxins used as antithrombotic drugs. Disintegrins act on the platelet surface via various receptors and glycoproteins, and fibrinolytic enzymes act by digesting fibrin to inhibit thrombus formation. ADP, adenosine diphosphate; BPPs, bradykinin-potentiating peptides; CTLs, C-type lectins, GPIb-V-IX, platelet glycoprotein adhesion receptor; GPIIb/IIIa, platelet-specific integrin receptor (α_IIb_β_3_); P2Y12-R, P2Y1-R, platelet G-protein-coupled receptors for ADP; PAR1/4-R, protease-activated receptors (G-protein-coupled receptors for thrombin); PLA_2_s, phospholipases A_2_; 3FTx, three-finger toxins; vWF: von Willebrand factor.

## Data Availability

Not applicable.

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
