# Peer review of "Snake Venom Components as Therapeutic Drugs in Ischemic Heart Disease"

_biomolecules, 2023, doi:10.3390/biom13101539_

Round 1
Reviewer 1 Report
This a very well written revision on the action of snake venom toxins and their derivatives on the cardiovascular system, and their use in the treatment of cardiovascular diseases. Congratulations.
I have noticed that it is missing C-type lectin-like proteins in Fig. 1. Their action on the cardiovascular system should be explained, and their potential use as anti-platelet drugs.
Minor comments:
Lines 89-92: review numeration.
Line 198: change yest to yet.
Line 219: change jaraca to jararaca.
Fig. 3 - the platelet has the color and shape of a red blood cell. The figure causes misleading interpretation. Edit the figure to become more self-explicative.
Author Response
Response to Reviewer 1 Comments
Comments and Suggestions for Authors
This a very well written revision on the action of snake venom toxins and their derivatives on the cardiovascular system, and their use in the treatment of cardiovascular diseases. Congratulations.
Comments (in black) and answers (in red):
Point 1: I have noticed that it is missing C-type lectin-like proteins in Fig. 1. Their action on the cardiovascular system should be explained, and their potential use as anti-platelet drugs.
Response 1: First, we would like to thank the reviewer for the time spent reading and editing this document, as well as for the very useful suggestions to improve the quality of this article. To address Point 1 raised by the reviewer, we have not discussed C-type lectins in the initial version of the review because, to our knowledge, C-type lectins (CTLs) have not been used as cardiovascular therapeutics to date. However, given the anti-platelet aggregation effects that some CTLs may have, we have added in the revised manuscript a paragraph (5. C-type lectin-like proteins ; pages 12-13) to discuss their action on the cardiovascular system and potential use as anti-platelet drugs. CTLs were also added Figures 1 and 3, and in Tables 1 and 2.
Point 2: Minor comments:
Lines 89-92: review numeration.
Line 198: change yest to yet.
Line 219: change jaraca to jararaca.
Fig. 3 - the platelet has the color and shape of a red blood cell. The figure causes misleading interpretation. Edit the figure to become more self-explicative.
Response 2: As recommended by the reviewer in Point 2, all the changes required have been performed. Figure 3 (with the new shape of platelets, as shown below) has been edited and inserted in the revised manuscript.
Point 3: Minor editing of English language required.
Response 3: English language has been edited as suggested by the reviewer, as well as typographical errors. All modifications have been inserted in the text of the revised manuscript.
Reviewer 2 Report
Manuscript no.: Biomolecules-2590879
Title: Snake venom components as therapeutic drugs in ischemic heart disease.
Author: E. Messadi
General comments:
In recent years, numerous reviews have addressed the cardiovascular actions of snake venoms and their toxins, and their potential use as lead molecules for the development of novel therapeutic compounds. In this review, the author discusses the potential usefulness of selected snake venom toxins [bradykinin-potentiating peptides – BPPs, natriuretic peptides – NPs, disintegrins, three-finger toxins (3FTx), snake venom VEGF, sarafotoxins, cysteine-rich secretory proteins (CRISPs), fibrinolytic enzymes (that include metallo- and serine proteinases) and phospholipases A2] for the treatment of ischemic heart disease (IHD) and related conditions. This review provides a general overview of the actions and applications of selected snake venoms toxins, although there is considerable variation in the amount and quality of the evidence that these toxins may indeed be useful therapeutically.
Although a considerable amount of the information summarized in this article has been dealt with to varying degrees in other reviews that have discussed the possible medicinal or therapeutic applications of snake venom toxins, in this work the focus is limited to the use of toxins in the study and treatment of IHD and conditions that may lead to it (atherosclerosis, hypertension, ischemia-reperfusion injury, etc.). While potentially interesting, there are several aspects of this work that are of concern and need to be addressed:
1. A major concern is that there is a considerable amount of inference and speculation on the potential uses of various venom components in IHD, but relatively few concrete studies or data with a direct bearing on this particular subject. Indeed, of the various toxins discussed, convincing (‘strong’) evidence for a possible/potential role in IHD and/or associated conditions (atherosclerosis, hypertension, ischemia-reperfusion injury, etc.) is limited primarily to BPPs, NPs, disintegrins and fibrinolytic enzymes – all of which have therapeutically relevant derivative molecules currently in use. In contrast, the evidence of a possible role for PLA2, 3FTx, CRISPs, svVEGF and sarafotoxins in IHD is limited and weak at best, with their inclusion here based on a foreseen potential, rather than a conclusive demonstration of relevance. This considerable variation in the range and quality of evidence needs to be made clear to the reader. Perhaps the author could establish criteria for classifying the toxins and their studies as, e.g., confirmed and applied (demonstrated by clinical studies and in therapeutic use), potentially useful (demonstrated in basic, experimental animal studies, but limited or no clinical evidence or trials), uncertain usefulness (very limited and not necessarily convincing data from animal studies, with no clinical studies). Such a classification would help the reader understand which compounds (or derivatives) are currently undergoing clinical investigation/trials and which offer a real perspective of becoming therapeutically relevant.
2. This is essentially a descriptive review. As such, the quality of this manuscript would be improved if there were a more critical analysis and assessment of the studies (experimental or clinical) currently in progress. For example, how closely do the animal models used resemble or reproduce the clinical situation in humans? A further point relates to inference from in vitro studies. There needs to be caution in extrapolating studies in vitro to possible in vivo use or applications since what is seen in the former is not always applicable to or work in the latter. Part of the problem here is related to the artificial conditions of most in vitro assays and the toxin concentrations tested in vitro which are often quite high and unlikely to be achieved in vivo.
3. The relevance of the article would also be enhanced by including a discussion of the challenges and limitations involved in developing therapeutic molecules from snake venom toxins. Such considerations would include pharmacological selectivity, pharmacokinetic aspects, and the difficulty in designing stable derivative molecules, especially from toxins of large molecular size, e.g., enzymes. It is no coincidence that many (if not most) of the therapeutically effective snake toxin-derived molecules currently in use are derived from or based on low molecular mass venom compounds, e.g., BPPs, NPs, and disintegrins, when compared to higher molecular mass components such as enzymes (a possible exception being fibrinolytic enzymes). Since snake venoms contain numerous non-BPP peptides that have a range of biological activities [see, for example, Villar-Briones A, Aird SD (2018) Organic and peptidyl constituents of snake venoms: the picture is vastly more complex than we imagined. Toxins 10, 392. Doi: 10.3390/toxins10100392], perhaps the author should devote some attention to these. See also these publications: (1) Xu X, Li B, Zhu S, Rong R (2015) Hypotensive peptides from snake venoms: structure, function and mechanism. Curr Top Med Chem 15, 658-669. Doi: 10.2174/1568026615666150217113835. (2) Almeida JR, Resende LM, Watanabe RK, Carregari VC, Huancahuire-Vega S, Caldeira CAS, Coutinho-Neto A, Soares AM, Vale N, Gomes PAC, Marangoni S, Calderon LS, Da Silva SL (2017) Snake venom peptides and low mass proteins: molecular tools and therapeutic agents. Curr Med Chem 24, 3254-3282. Doi: 10.2174/0929867323666161028155611.
4. For toxins such as PLA2 that have shown little applicability in IHD to date, one possibility would be to investigate the usefulness of biologically active fragments of these larger molecules – from the C-terminal region in the case of PLA2 (as shown for antimicrobial and other activities). Indeed, since most therapeutically relevant compounds for IDH and other cardiovascular conditions are low molecular mass molecules, this is likely to be a more promising approach than to develop a product based on the entire toxin; such an approach could also reduce potential adverse effects associated with larger molecules. This matter requires some comment.
5. Several of the toxins discussed in this work, e.g., NPs, VEGF, sarafotoxins have equivalent natural (endogenous) counterparts in human physiology. In view of this, there needs to be some comment on why these snake toxins should be used as lead molecules for producing novel therapeutic compounds rather than the endogenous molecules themselves. What advantages/disadvantages do they offer compared to the endogenous molecules?
Specific comments:
1. Line 112: What is meant by the phrase ‘anti-reactive oxygen species’? What chemical species or compounds are these?
2. BPPs:
a) There should be some comment on whether BPPs show similar affinity for and inhibition of ACE1 and ACE2. For discussion of a possible interaction of BPPs and ACE2, see: Gouda AS, Mégarbane B (2021) Snake venom-derived bradykinin-potentiating peptides: A promising therapy for COVID-19? Drug Dev Res 82, 38-48. Doi: 10.1002/ddr.21732.
b) The logic in lines 137-150 is unclear. Are BPPs equally active at both domains (N and C)? What is the importance or relevance of N-domain blockade by captopril? If the C-domain is necessary for BK hydrolysis (and its blockade leads to angioedema), then what is the point of using Azemiops feae BPPs to develop more C-domain-specific inhibitors?
c) For further information on the structure-function relationships of BPPs, the author should consult Sciani JM, Pimenta DC (2017) The modular nature of bradykinin-potentiating peptides isolated from snake venoms. J Venom Anim Toxins Incl Trop Dis 23, 45. Doi: 10.1186/s40409-017-0134-7, and also a chapter in the volume edited by S.P. Mackessy (ref. 144 in the manuscript list)
d) The discussion in sections 2.1 to 2.3 is highly speculative and inferential. Much of sections 2.1 and 2.3 is based on captopril and other commercial ACE inhibitors and not on BPPs themselves and, as such, is not directly relevant to the subject (which is ‘snake venom components’). In section 2.2, while it is true that a variety of non-ACE-mediated effects have been described for BPPs, most of these studies were done a decade or more ago and are no longer being continued since the head of the research group involved in most of these studies has since retired and there is no continuation of this investigation by pharmaceutical companies. In view of these considerations, this section on BPPs should focus strictly on studies in which these peptides (and not their derivatives such as captopril, etc.) have definitely been shown to be beneficial in IHD, ischemia-reperfusion injury, hypertension and atherosclerosis. Avoid speculation and inference.
3. Many of the references in the section on sarafotoxins are quite old (only two, refs. 192 and 193, are <10 years old). A more up-to-date literature search needs to be done to adequately assess whether there have been any recent advances in cardiovascular (ischemic heart disease, etc.) studies with these toxins in recent years. A lack of recent studies may indicate that this is not a particularly productive topic for investigation.
4. In view of the lack of solid evidence for the applicability of PLA2, 3FTx, CRISPs, svVEGF and sarafotoxins in IHD and related conditions, it would be better to be less speculative about these toxin groups and considerably reduce the extent of discussion of these toxins, which currently occupies pages 13-19 of the manuscript.
5. Lines 468-469: The sentence ‘These findings in aorta...and thoracic aorta.’ is unclear and somewhat redundant. Please clarify
6. Line 669: What is meant by the phrase ‘more or less different effects’?
Other issues:
1. Place keywords in alphabetical order.
2. Avoid use of the form ‘we’ as this work has only one author. Better to say: This review describes the snake venom components that interact with the CVS... (lines 92 and 93).
3. Standardize the abbreviation for three-finger toxins to ‘3FTx’ throughout the text; this is a widely used abbreviation in the literature; avoid the use of ‘TFT’, even though it has been used in some publications.
4. Throughout the main text, tables, figures and legends and references, place the ‘2’ of PLA2 in subscript.
5. Sections 2.1, 2.2 and 2.3 – Correct ‘BBPs’ to read ‘BPPs’ in section titles and text.
6. Line 219: Correct to read ‘Bothrops jararaca’
7. Line 249: ...originating Dendroaspis NP... (place Dendroaspis in italics as it is a genus name)
8. Place the ‘ATP’ in ‘KATP’ in subscript throughout the text, figures, tables and legends.
Figures and Table
Figure 1
1. In the figure, remove the ‘s’ from ‘proteins’ and ‘peptides’, e.g., Protein and peptide components.
2. Legend: Correct to read ‘Main snake venom components with....
Figure 2
1. In VSMC, add ‘channels’ to read ‘L-type Ca2+ channels’
2. In the figure and legend, place the ‘2’ of ‘PGI2’ in subscript. Also for ‘PLA2’
Figure 3
1. Correct the various abbreviations (BPPs – not BBPs; PLA2 with subscript ‘2’, 3FTX rather than TFT, etc.) as already indicated.
Table 1:
1. The column for ‘Molecular weight (kDa)’ should read ‘Molecular mass (kDa)’
2. The molecular mass range for BPPs should be ~500-3.0 rather than 1.5-3.0
3. The receptor types for sarafotoxins should be ETA and ETB (not ET(A), ET(B))
4. For alternagin-C, fourth column, correct to read ‘reoxygenation’ (without accents)
References:
The reference list needs to be checked very carefully to standardize the information provided and to align the style and formatting with the journal´s instructions. Specifically, these alterations include (but are not limited to):
General
1. List all authors and use the correct format and order for initials and surnames. Do not use ‘and’ between penultimate and last author, e.g., refs. 2,3,7 and others.
2. Do not capitalize the first letter of each major word in a title, e.g., refs. 2,7,8 and others
3. Provide the journal´s name that is missing in various cases, e.g., refs. 2,7,11 and others.
4. Provide complete page numbers where available, e.g., 1234-1239, not 1234-9, etc. Applies to many references. For online or open access journals, provide at least the first page number, e.g., ref. 125,150 and others.
5. Provide full-stops (or periods) after abbreviated journal names, e.g., Curr. Med. Chem., not Curr Med Chem.
6. Place publication year in bold type and volume number in italics; no need for journal part (fascicle) numbers; use correct punctuation for journal citations, e.g., commas between year and volume number and between volume number and page numbers
7. Ensure that Greek letters are used in article titles where appropriate, e.g., refs. 161-163 – should the ‘alpha’ and ‘beta’ in these titles be in Greek letters rather than being spelt out? Check for other similar cases.
Specific
1. Reference 1: Correct to read ‘World Health Organization’ (not: Organization, W.H.), and provide the place of publication.
2. Reference 4: Correct the author´s name; do not capitalize journal name.
3. References 10 and 155: Correct to read ‘FASEB J’.
4. Reference 60: Place the ’2’ in H2O2 in subscript
5. Reference 64: Correct to read ‘New’, with capital ‘N’
6. References 80 and 81 are repeated; delete one.
7. References 98,146,147,150: Abbreviate journal names
8. References 121 and 122: Provide authors´ names
9. References 144 and 201: Indicate that Mackessy, S.P. is the editor of these two volumes. Also indicate whether you are referring to the first or second editions of publications and provide the place of publication. Since these are edited publications, it would be better to indicate the specific chapters to which you are referring.
10. Reference 147: Author´s name (HINTZE) should be in lowercase (Hintze)
11. Reference 176: Complete the bibliographic data – journal name, volume, page numbers, etc.
12. Reference 189: Place ‘2+’ in ‘Ca2+’ in superscript.
13. Reference 206: Correct to read ‘Philippe, F.’...
Grammar:
The entire manuscript would benefit from rigorous proofreading and editing to improve the quality of the text. A few examples are given below, but there are many others in the text.
Line Correction
11 ...size, this intervention has nevertheless been shown to exacerbate...
13 ‘that’ instead of ‘which’: ...and peptides that are of major...
16-17 ...developed from a peptide present in Bothrops jararaca snake venom. This review discusses the potential usefulness of snake venom toxins for developing ... Place species name in italics.
20 The molecules described here have either...and are currently...market or are still in clinical...
22-23 ...stages... The information summarized here may be useful in providing insights....
28 ...affecting the cardiovascular system (CVS) and analyses the...
34 ...disease, reaching 9 million [1,2]....
36 ...one of the main manifestations and causes of death from IHD, with other manifestations including stable angina...
41 ...build-up of atherosclerotic plaque in coronary arteries [4,5].
45 ...that has been termed
46 ...date, there has been considerable...
50 ...IHD is still the...
51 ...the need to identify and develop new therapeutic...
52 Delete the phrase: ...which are major issues...IHD occurrence.
56 Use the term ‘envenoming’ instead of ‘poisoning’
71 ...drive the discovery...
72 ...and many others may appear as studies continue to investigate snake venom...
76 ...venom dry weight...
78 These high affinity, selective compounds [18] can participate in a wide range of physiological events through their individual activities or through synergy with other molecules [23].
91 Point (ii) here should be (iii).
368 ...such as alternagin-C, from Bothrops alternatus venom, which has been found...
402 ...enzymes involved in fibrinolysis (Figure 3)...
Grammar:
The entire manuscript would benefit from rigorous proofreading and editing to improve the quality of the text. A few examples are given below, but there are many others in the text.
Line Correction
11 ...size, this intervention has nevertheless been shown to exacerbate...
13 ‘that’ instead of ‘which’: ...and peptides that are of major...
16-17 ...developed from a peptide present in Bothrops jararaca snake venom. This review discusses the potential usefulness of snake venom toxins for developing ... Place species name in italics.
20 The molecules described here have either...and are currently...market or are still in clinical...
22-23 ...stages... The information summarized here may be useful in providing insights....
28 ...affecting the cardiovascular system (CVS) and analyses the...
34 ...disease, reaching 9 million [1,2]....
36 ...one of the main manifestations and causes of death from IHD, with other manifestations including stable angina...
41 ...build-up of atherosclerotic plaque in coronary arteries [4,5].
45 ...that has been termed
46 ...date, there has been considerable...
50 ...IHD is still the...
51 ...the need to identify and develop new therapeutic...
52 Delete the phrase: ...which are major issues...IHD occurrence.
56 Use the term ‘envenoming’ instead of ‘poisoning’
71 ...drive the discovery...
72 ...and many others may appear as studies continue to investigate snake venom...
76 ...venom dry weight...
78 These high affinity, selective compounds [18] can participate in a wide range of physiological events through their individual activities or through synergy with other molecules [23].
91 Point (ii) here should be (iii).
368 ...such as alternagin-C, from Bothrops alternatus venom, which has been found...
402 ...enzymes involved in fibrinolysis (Figure 3)...
Author Response
Response to Reviewer 2 Comments
Comments and Suggestions for Authors
Manuscript no.: Biomolecules-2590879
Title: Snake venom components as therapeutic drugs in ischemic heart disease.
Author: E. Messadi
General comments:
In recent years, numerous reviews have addressed the cardiovascular actions of snake venoms and their toxins, and their potential use as lead molecules for the development of novel therapeutic compounds. In this review, the author discusses the potential usefulness of selected snake venom toxins [bradykinin-potentiating peptides – BPPs, natriuretic peptides – NPs, disintegrins, three-finger toxins (3FTx), snake venom VEGF, sarafotoxins, cysteine-rich secretory proteins (CRISPs), fibrinolytic enzymes (that include metallo- and serine proteinases) and phospholipases A2] for the treatment of ischemic heart disease (IHD) and related conditions. This review provides a general overview of the actions and applications of selected snake venoms toxins, although there is considerable variation in the amount and quality of the evidence that these toxins may indeed be useful therapeutically.
Although a considerable amount of the information summarized in this article has been dealt with to varying degrees in other reviews that have discussed the possible medicinal or therapeutic applications of snake venom toxins, in this work the focus is limited to the use of toxins in the study and treatment of IHD and conditions that may lead to it (atherosclerosis, hypertension, ischemia-reperfusion injury, etc.). While potentially interesting, there are several aspects of this work that are of concern and need to be addressed:
Point 1: A major concern is that there is a considerable amount of inference and speculation on the potential uses of various venom components in IHD, but relatively few concrete studies or data with a direct bearing on this particular subject. Indeed, of the various toxins discussed, convincing (‘strong’) evidence for a possible/potential role in IHD and/or associated conditions (atherosclerosis, hypertension, ischemia-reperfusion injury, etc.) is limited primarily to BPPs, NPs, disintegrins and fibrinolytic enzymes – all of which have therapeutically relevant derivative molecules currently in use. In contrast, the evidence of a possible role for PLA2, 3FTx, CRISPs, svVEGF and sarafotoxins in IHD is limited and weak at best, with their inclusion here based on a foreseen potential, rather than a conclusive demonstration of relevance. This considerable variation in the range and quality of evidence needs to be made clear to the reader. Perhaps the author could establish criteria for classifying the toxins and their studies as, e.g., confirmed and applied (demonstrated by clinical studies and in therapeutic use), potentially useful (demonstrated in basic, experimental animal studies, but limited or no clinical evidence or trials), uncertain usefulness (very limited and not necessarily convincing data from animal studies, with no clinical studies). Such a classification would help the reader understand which compounds (or derivatives) are currently undergoing clinical investigation/trials and which offer a real perspective of becoming therapeutically relevant.
Response 1: First of all, I would like to thank reviewer #2 for the time spent evaluating my work and for the valuable comments. I hope that my answers and clarifications will help improve the quality of the manuscript. To address the comment raised by the reviewer, I added and inserted in the new revised version of the article a new Table (Table 2), at the end of the "Introduction" section, where the toxins mentioned in the work are now classified according to their usefulness in cardiovascular therapy. As shown below, Table 1 includes 4 sections : 1/Approved snake-venom-based drugs, 2/ Orphan snake venom-based drugs, 3/ Toxins of interest in experimental and pre-clinical studies and 4/ Toxins with limited experimental data. Furthermore, for more clarity, some other changes were made in the revised manuscript in particular by presenting the toxins throughout the text according to their non-enzymatic activity first (Sections 2 to 9) and then enzymatic properties (Sections 10 & 11).
Table 2. Classification of main snake venom toxins according to their usefulness in cardiovascular therapy or research.
|
Approved snake-venom-based drugs |
||||
|
Peptide/Protein |
Origin |
Drug |
Mode of action |
Cardiovascular indications |
|
BPP5a, BPP9a (Bradykinin potentiating peptides, BPPs) |
Bothrops jararaca |
Captopril/ Enalapril |
ACE inhibitors |
Hypertension, MI, HF [1]. |
|
Barbourin (Disintegrin) |
Sistrurus m. barbouri |
Integrilin/ Eptifibatide |
GPIIb/IIIa antagonist |
ACS and anti-thrombotic therapy [2, 3]. |
|
Echistatin (Disintegrin) |
Echis carinatus |
Aggrastat/ Tirofiban |
GPIIb/IIIa antagonist |
ACS and anti-thrombotic therapy [2, 3]. |
|
Batroxobin (Fibrinolytic enzyme) |
Bothrops moojeni |
Defibrase |
Cleavage of fibrinogen Aα subunit |
Anticoagulant therapy in ACS [4, 5]. |
|
Orphan snake venom-based drugs |
||||
|
Peptide/Protein |
Origin |
Drug |
Mode of action |
Cardiovascular indications |
|
DNP (Natriuretic peptide) |
Dendroaspis angusticeps |
Cenderitide (CD-NP) |
NPR-A and -B agonist |
HF [6, 7]. |
|
Fibrolase (Fibrinolytic enzyme)
|
Agkistrodon contortrix contortrix |
Alfimeprase |
Cleavage of fibrinogen α- and β-chains |
Acute ischemic stroke, acute peripheral arterial occlusion, catheter occlusion [8-10]. |
|
Ancrod (Fibrinolytic enzyme) |
Calloselasma rhodostoma |
Viprinex |
Cleavage of fibrinogen α-chain |
Anticoagulant therapy in thrombosis [9, 11]. |
|
Toxins of interest in experimental and pre-clinical studies |
||||
|
Peptide/Protein family |
Origin |
Molecule |
Mode of action |
Cardiovascular indications |
|
Natriuretic peptides |
Macrovipera lebetina |
Lebetin 2 (L2) |
NPR-A and -B agonist |
Increased BP, MI [12-14]. |
|
Disintegrins |
Rhinocerophis alternates |
Alternagin-C |
GPIa/IIa (α2β1 integrin) antagonist VEGFR-2 inhibition |
MI [15, 16]. |
|
C-type Lectins (CTLs) |
Vipera palestinae |
Vipegitide |
GPIa/IIa (α2β1 integrin) antagonist |
Antithrombotic therapy [2, 17]. |
|
Three-finger toxins (3FTx) |
Dendroaspis jamesoni kaimosae |
Dendroaspin (mambin) |
GPIIb/IIIa (Integrin αIIbβ3) antagonist |
Antithrombotic therapy [18]. |
|
Vascular endothelial growth factors (svVEGFv) |
Macrovipera lebetina |
ICPP |
VEGF-A agonist |
Increased BP, acute MI [19]. |
|
Sarafotoxins (SRTXs) |
Atractaspis engaddensis |
SRTX-6c |
Endothelin receptors ET(B) agonist |
MI [20-23]. |
|
Toxins with limited experimental data |
||||
|
Peptide/Protein family |
Origin |
Molecule |
Mode of action |
Cardiovascular indications |
|
Cysteine-rich secretory proteins (CRISPs) |
Gloydius blomhoffii |
Ablomin |
L-type voltage-gated Ca2+ channel blockade |
Hypertension [24, 25]. |
|
Phospholipases A2 (PLA2s) |
Oxyuranus scutellatus Vipera palestinae |
OSC3 PLA2 isoforms |
Hypertension [26]. Atherosclerosis [27]. |
|
ACE, angiotensin-converting enzyme ; ACS, acute coronary syndrome ; DNP, Dendroaspis natriuretic peptide ; HF, heart failure ; ICPP, increasing capillary permeability protein ; MI, myocardial infarction, NPR-A and -B, natriuretic peptide receptors A and B ; VEGR-2, Vascular endothelial growth factor receptors 2.
Point 2: This is essentially a descriptive review. As such, the quality of this manuscript would be improved if there were a more critical analysis and assessment of the studies (experimental or clinical) currently in progress. For example, how closely do the animal models used resemble or reproduce the clinical situation in humans? A further point relates to inference from in vitro studies. There needs to be caution in extrapolating studies in vitro to possible in vivo use or applications since what is seen in the former is not always applicable to or work in the latter. Part of the problem here is related to the artificial conditions of most in vitro assays and the toxin concentrations tested in vitro which are often quite high and unlikely to be achieved in vivo.
Response 2: I thank the reviewer for the relevance of this comment which I fully agree. Due to the lack of complete physiological environment and neurohumoral connections, in vitro data should be extrapolated with caution to in vivo setting. Caution should also be exercised when extrapolating animal data for potential use in human therapeutics, as several venom compounds showing promising results in animal studies, have not been approved in subsequent clinical trials (Table 2). This comment was addressed in the “Introduction” section (above Table 2) of the revised manuscript for more critical assessment of the review. Furthermore, Table 2 added in the revised manuscript provides clarification regarding this information.
Point 3: The relevance of the article would also be enhanced by including a discussion of the challenges and limitations involved in developing therapeutic molecules from snake venom toxins. Such considerations would include pharmacological selectivity, pharmacokinetic aspects, and the difficulty in designing stable derivative molecules, especially from toxins of large molecular size, e.g., enzymes. It is no coincidence that many (if not most) of the therapeutically effective snake toxin-derived molecules currently in use are derived from or based on low molecular mass venom compounds, e.g., BPPs, NPs, and disintegrins, when compared to higher molecular mass components such as enzymes (a possible exception being fibrinolytic enzymes). Since snake venoms contain numerous non-BPP peptides that have a range of biological activities [see, for example, Villar-Briones A, Aird SD (2018) Organic and peptidyl constituents of snake venoms: the picture is vastly more complex than we imagined. Toxins 10, 392. Doi: 10.3390/toxins10100392], perhaps the author should devote some attention to these. See also these publications: (1) Xu X, Li B, Zhu S, Rong R (2015) Hypotensive peptides from snake venoms: structure, function and mechanism. Curr Top Med Chem 15, 658-669. Doi: 10.2174/1568026615666150217113835. (2) Almeida JR, Resende LM, Watanabe RK, Carregari VC, Huancahuire-Vega S, Caldeira CAS, Coutinho-Neto A, Soares AM, Vale N, Gomes PAC, Marangoni S, Calderon LS, Da Silva SL (2017) Snake venom peptides and low mass proteins: molecular tools and therapeutic agents. Curr Med Chem 24, 3254-3282. Doi: 10.2174/0929867323666161028155611.
Response 3: I thank the reviewer for this insightful comment. I have added a “Discussion” section in the revised manuscript (before the “Conclusion” section) to discuss drawbacks, limitations and challenges in the field of snake venom toxinology (Section 12, pages 17-18)
Point 4: For toxins such as PLA2 that have shown little applicability in IHD to date, one possibility would be to investigate the usefulness of biologically active fragments of these larger molecules – from the C-terminal region in the case of PLA2 (as shown for antimicrobial and other activities). Indeed, since most therapeutically relevant compounds for IDH and other cardiovascular conditions are low molecular mass molecules, this is likely to be a more promising approach than to develop a product based on the entire toxin; such an approach could also reduce potential adverse effects associated with larger molecules. This matter requires some comment.
Response 4: The approach mentioned by the reviewer is very relevant, given that PLA2 could be a promising molecule in ischemic heart disease, particularly in terms of anti-hypertensive and potentially anti-atheroscerotic effects. As suggested by the reviewer, this comment was addressed and discussed in the “Phosphilpases A2” section (Section 11 of the revised manuscript ; above the “Discussion” section).
Point 5: Several of the toxins discussed in this work, e.g., NPs, VEGF, sarafotoxins have equivalent natural (endogenous) counterparts in human physiology. In view of this, there needs to be some comment on why these snake toxins should be used as lead molecules for producing novel therapeutic compounds rather than the endogenous molecules themselves. What advantages/disadvantages do they offer compared to the endogenous molecules?
Response 5: Although several venom molecules, such as BPPs, NPs and VEGFs, have equivalent endogenous counterparts in human physiology, it is more relevant to use them rather than natural molecules in certain situations. This is the case when human recombinant proteins cause unexpected adverse effects, leading to the use of venom compounds. This was particularly observed with neseritide (a recombinant human BNP used as a drug for HF treatment) which induces deleterious hypotension in treated patients. To address this issue, Cenderitide was designed from venom toxin DNP, in order to compensate hypotension induced by endogenous NPs, in particular by fusing it to CNP. Second, venom molecules may be more attractive to use than their human counterparts when they exhibit more potent and/or additional effects, usually involving different (and/or novel) therapeutic targets. For example, we discovered, in our work, that L2, a snake venom NP, exerts more potent and additional effects that are not observed with human BNP, involving different therapeutic targets and pathways, comapred to human NPs (Tourki et al., 2016; Tourki et al., 2019, Allaoui et al., 2022; Bouzazi, et al., 2023). Another example is lebecetin, a potentail C-type lectin-based drug (currently patented), for its high selectivity and potent ability to induce multiple cellular targets, which makes it more potent than other synthetic family members in the treatment of neovascular eye diseases (Montassar et al., 2017). Due to their ability to escape human enzymatic systems, leading to higher stability, some snake venom toxins are useful as diagnostic tools, e.g. for the assay of coagulation factors and for the study of haemostasis (Marsh, NA., 2002).
Some comments relating to this topic have been inserted in the “Conclusion” section.
Point 6: Specific comments:
- Line 112: What is meant by the phrase ‘anti-reactive oxygen species’? What chemical species or compounds are these?
Bradykinin (BK) has been proven to protect cardiomyocytes against the effect of oxidative stress through the activation of the BK receptor B2 (B2-R) which induces upregulation of antioxidant Cu/Zn-superoxide dismutase (SOD) and Mn-SOD activity and expression, and downregulation of NADPH oxidase activity, and subsequent inhibition of reactive oxygen species (ROS) production (Dong et al., 2013)à reference added in the text
- BPPs:
- a)There should be some comment on whether BPPs show similar affinity for and inhibition of ACE1 and ACE2. For discussion of a possible interaction of BPPs and ACE2, see: Gouda AS, Mégarbane B (2021) Snake venom-derived bradykinin-potentiating peptides: A promising therapy for COVID-19? Drug Dev Res 82, 38-48. Doi: 10.1002/ddr.21732.
- b)The logic in lines 137-150 is unclear. Are BPPs equally active at both domains (N and C)? What is the importance or relevance of N-domain blockade by captopril? If the C-domain is necessary for BK hydrolysis (and its blockade leads to angioedema), then what is the point of using Azemiops feae BPPs to develop more C-domain-specific inhibitors?
Response (b): This comment was addressed in the text (Section 2.BPPs, page 7 of the revised manuscript).
- c)For further information on the structure-function relationships of BPPs, the author should consult Sciani JM, Pimenta DC (2017) The modular nature of bradykinin-potentiating peptides isolated from snake venoms.J Venom Anim Toxins Incl Trop Dis 23, 45. Doi: 10.1186/s40409-017-0134-7, and also a chapter in the volume edited by S.P. Mackessy (ref. 144 in the manuscript list)
Response (c): This reference was inserted in the revised manuscript_Reference #[95].
- d)The discussion in sections 2.1 to 2.3 is highly speculative and inferential. Much of sections 2.1 and 2.3 is based on captopril and other commercial ACE inhibitors and not on BPPs themselves and, as such, is not directly relevant to the subject (which is ‘snake venom components’). In section 2.2, while it is true that a variety of non-ACE-mediated effects have been described for BPPs, most of these studies were done a decade or more ago and are no longer being continued since the head of the research group involved in most of these studies has since retired and there is no continuation of this investigation by pharmaceutical companies. In view of these considerations, this section on BPPs should focus strictly on studies in which these peptides (and not their derivatives such as captopril, etc.) have definitely been shown to be beneficial in IHD, ischemia-reperfusion injury, hypertension and atherosclerosis. Avoid speculation and inference.
Response (d): The text of these sections has been modified as suggested by the reviewer.
- Many of the references in the section on sarafotoxins are quite old (only two, refs. 192 and 193, are <10 years old). A more up-to-date literature search needs to be done to adequately assess whether there have been any recent advances in cardiovascular (ischemic heart disease, etc.) studies with these toxins in recent years. A lack of recent studies may indicate that this is not a particularly productive topic for investigation.
- In view of the lack of solid evidence for the applicability of PLA2, 3FTx, CRISPs, svVEGF and sarafotoxins in IHD and related conditions, it would be better to be less speculative about these toxin groups and considerably reduce the extent of discussion of these toxins, which currently occupies pages 13-19 of the manuscript.
- Lines 468-469: The sentence ‘These findings in aorta...and thoracic aorta.’ is unclear and somewhat redundant. Please clarify
- Line 669: What is meant by the phrase ‘more or less different effects’?
Response 6: The responses to the comments mentioned in Point 6 (1 to 6) have been addressed as possible, (and for some indicated in red under the comment subsections) ; modifications also have been inserted in the revised text in relation to the reviewer’s comments in this section.
Point 7: Other issues:
- Place keywords in alphabetical order.
- Avoid use of the form ‘we’ as this work has only one author. Better to say: This review describes the snake venom components that interact with the CVS... (lines 92 and 93).
- Standardize the abbreviation for three-finger toxins to ‘3FTx’ throughout the text; this is a widely used abbreviation in the literature; avoid the use of ‘TFT’, even though it has been used in some publications.
- Throughout the main text, tables, figures and legends and references, place the ‘2’ of PLA2 in subscript.
- Sections 2.1, 2.2 and 2.3 – Correct ‘BBPs’ to read ‘BPPs’ in section titles and text.
- Line 219: Correct to read ‘Bothrops jararaca’
- Line 249: ...originating Dendroaspis NP... (place Dendroaspis in italics as it is a genus name)
- Place the ‘ATP’ in ‘KATP’ in subscript throughout the text, figures, tables and legends.
Response 7: For the “Other issues” comments, all the points from 1 to 8 have been corrected and modified in the revised manuscript (text, legend, graphs and tables)
Point 8: Figures and Table
Figure 1
- In the figure, remove the ‘s’ from ‘proteins’ and ‘peptides’, e.g., Protein and peptide components.
- Legend: Correct to read ‘Main snake venom components with....
Figure 2
- In VSMC, add ‘channels’ to read ‘L-type Ca2+ channels’
- In the figure and legend, place the ‘2’ of ‘PGI2’ in subscript. Also for ‘PLA2’
Figure 3
- Correct the various abbreviations (BPPs – not BBPs; PLA2 with subscript ‘2’, 3FTX rather than TFT, etc.) as already indicated.
Table 1:
- The column for ‘Molecular weight (kDa)’ should read ‘Molecular mass (kDa)’
- The molecular mass range for BPPs should be ~500-3.0 rather than 1.5-3.0
- The receptor types for sarafotoxins should be ETA and ETB (not ET(A), ET(B))
- For alternagin-C, fourth column, correct to read ‘reoxygenation’ (without accents)
Response 8: For the “Figures and Tables” comments, all the points have been corrected and modified in the revised manuscript (graphs, legend and tables)
Point 9: References:
The reference list needs to be checked very carefully to standardize the information provided and to align the style and formatting with the journal´s instructions. Specifically, these alterations include (but are not limited to):
General
- List all authors and use the correct format and order for initials and surnames. Do not use ‘and’ between penultimate and last author, e.g., refs. 2,3,7 and others.
- Do not capitalize the first letter of each major word in a title, e.g., refs. 2,7,8 and others
- Provide the journal´s name that is missing in various cases, e.g., refs. 2,7,11 and others.
- Provide complete page numbers where available, e.g., 1234-1239, not 1234-9, etc. Applies to many references. For online or open access journals, provide at least the first page number, e.g., ref. 125,150 and others.
- Provide full-stops (or periods) after abbreviated journal names, e.g., Curr. Med. Chem., not Curr Med Chem.
- Place publication year in bold type and volume number in italics; no need for journal part (fascicle) numbers; use correct punctuation for journal citations, e.g., commas between year and volume number and between volume number and page numbers
- Ensure that Greek letters are used in article titles where appropriate, e.g., refs. 161-163 – should the ‘alpha’ and ‘beta’ in these titles be in Greek letters rather than being spelt out? Check for other similar cases.
Specific
- Reference 1: Correct to read ‘World Health Organization’ (not: Organization, W.H.), and provide the place of publication.
- Reference 4: Correct the author´s name; do not capitalize journal name.
- References 10 and 155: Correct to read ‘FASEB J’.
- Reference 60: Place the ’2’ in H2O2 in subscript
- Reference 64: Correct to read ‘New’, with capital ‘N’
- References 80 and 81 are repeated; delete one.
- References 98,146,147,150: Abbreviate journal names
- References 121 and 122: Provide authors´ names
- References 144 and 201: Indicate that Mackessy, S.P. is the editor of these two volumes. Also indicate whether you are referring to the first or second editions of publications and provide the place of publication. Since these are edited publications, it would be better to indicate the specific chapters to which you are referring.
- Reference 147: Author´s name (HINTZE) should be in lowercase (Hintze)
- Reference 176: Complete the bibliographic data – journal name, volume, page numbers, etc.
- Reference 189: Place ‘2+’ in ‘Ca2+’ in superscript.
- Reference 206: Correct to read ‘Philippe, F.’...
Response 9: For General and specific “References” comments, all the checks have been made and modified in the revised manuscript (graphs, legend and tables)
Point 10: Grammar:
The entire manuscript would benefit from rigorous proofreading and editing to improve the quality of the text. A few examples are given below, but there are many others in the text.
Line Correction
11 ...size, this intervention has nevertheless been shown to exacerbate...
13 ‘that’ instead of ‘which’: ...and peptides that are of major...
16-17 ...developed from a peptide present in Bothrops jararaca snake venom. This review discusses the potential usefulness of snake venom toxins for developing ... Place species name in italics.
20 The molecules described here have either...and are currently...market or are still in clinical...
22-23 ...stages... The information summarized here may be useful in providing insights....
28 ...affecting the cardiovascular system (CVS) and analyses the...
34 ...disease, reaching 9 million [1,2]....
36 ...one of the main manifestations and causes of death from IHD, with other manifestations including stable angina...
41 ...build-up of atherosclerotic plaque in coronary arteries [4,5].
45 ...that has been termed
46 ...date, there has been considerable...
50 ...IHD is still the...
51 ...the need to identify and develop new therapeutic...
52 Delete the phrase: ...which are major issues...IHD occurrence.
56 Use the term ‘envenoming’ instead of ‘poisoning’
71 ...drive the discovery...
72 ...and many others may appear as studies continue to investigate snake venom...
76 ...venom dry weight...
78 These high affinity, selective compounds [18] can participate in a wide range of physiological events through their individual activities or through synergy with other molecules [23].
91 Point (ii) here should be (iii).
368 ...such as alternagin-C, from Bothrops alternatus venom, which has been found...
402 ...enzymes involved in fibrinolysis (Figure 3)...
Comments on the Quality of English Language
Grammar:
The entire manuscript would benefit from rigorous proofreading and editing to improve the quality of the text. A few examples are given below, but there are many others in the text.
Line Correction
11 ...size, this intervention has nevertheless been shown to exacerbate...
13 ‘that’ instead of ‘which’: ...and peptides that are of major...
16-17 ...developed from a peptide present in Bothrops jararaca snake venom. This review discusses the potential usefulness of snake venom toxins for developing ... Place species name in italics.
20 The molecules described here have either...and are currently...market or are still in clinical...
22-23 ...stages... The information summarized here may be useful in providing insights....
28 ...affecting the cardiovascular system (CVS) and analyses the...
34 ...disease, reaching 9 million [1,2]....
36 ...one of the main manifestations and causes of death from IHD, with other manifestations including stable angina...
41 ...build-up of atherosclerotic plaque in coronary arteries [4,5].
45 ...that has been termed
46 ...date, there has been considerable...
50 ...IHD is still the...
51 ...the need to identify and develop new therapeutic...
52 Delete the phrase: ...which are major issues...IHD occurrence.
56 Use the term ‘envenoming’ instead of ‘poisoning’
71 ...drive the discovery...
72 ...and many others may appear as studies continue to investigate snake venom...
76 ...venom dry weight...
78 These high affinity, selective compounds [18] can participate in a wide range of physiological events through their individual activities or through synergy with other molecules [23].
91 Point (ii) here should be (iii).
368 ...such as alternagin-C, from Bothrops alternatus venom, which has been found...
402 ...enzymes involved in fibrinolysis (Figure 3)...
Response 10: For “Grammar” comments, all the checks have been made and modified in the revised manuscript.
Reviewer 3 Report
There is currently a growing interest in the treatment of cardiovascular diseases on the basis of natural compounds. In this review, the authors give an overview on therapeutic applications based on various components based on snake venom. They discuss a broad panorama of components (proteins, peptides and others) derived from snake venom toxins.
I find this an interesting and stimulating paper, and it is basically well written. Therefore, I think it is suitable for publication.
Minor Comments
- It would be helpful if the terms used in lines 89-91 would correspond more closely to the terms in the headings of Table 1.
- It would increase readability if the authors would explicitly explain the structure of the paper (maybe at the end of the Introduction). Now the reader has to discover that the sections correspond to the proteins and peptides in Table 1. However, the order of the sections does not completely correspond to the order in Table 1, which is confusing.
- It would be good to be consistent in the term “Vascular Endothelial Growth Factor” for VEGF, and not call it “Endothelial vascular growth factors” in Table 1.
- Figures 2 and 3 are too big, and contain a lot of detail. Moreover, these figures are placed far from their explanations, which hampers understanding them. It would the good if the authors could address these problems.
- line 248: incudes -> includes
Author Response
Response to Reviewer 3 Comments
Comments and Suggestions for Authors
There is currently a growing interest in the treatment of cardiovascular diseases on the basis of natural compounds. In this review, the authors give an overview on therapeutic applications based on various components based on snake venom. They discuss a broad panorama of components (proteins, peptides and others) derived from snake venom toxins.
I find this an interesting and stimulating paper, and it is basically well written. Therefore, I think it is suitable for publication.
Minor Comments
Comments (in black) and answers (in red):
Point 1: It would be helpful if the terms used in lines 89-91 would correspond more closely to the terms in the headings of Table 1.
Response 1: First of all, I thank the reviewer for the time spent reading and editing this article, as well as for this helpful comment. For this first point, modifications have been made to the text (lines 70-77 of the revised manuscript) and to the Table 1 (the text and Table 1 are now homogeneous according to the reviewer's suggestion).
Point 2: It would increase readability if the authors would explicitly explain the structure of the paper (maybe at the end of the Introduction). Now the reader has to discover that the sections correspond to the proteins and peptides in Table 1. However, the order of the sections does not completely correspond to the order in Table 1, which is confusing.
Response 2: I agree with the reviewer's suggestion and have modified the structure of the article according to the order of toxins in Table 1. The manuscript now presents peptide and protein components without enzymatic activity first (Sections 2 to 9); then venom compounds with enzymatic activity (Sections 10 & 11). Moreover, another Table (Table 2, with 4 sections) was added at the end of the "Introduction" section, to classify the molecules presented in the work according to their usefulness in cardiovascular therapy (1/Approved snake-venom-based drugs, 2/ Orphan snake venom-based drugs, 3/ Toxins of interest in experimental and pre-clinical studies and 4/ Toxins with limited experimental data). The text was also modified to include these details for more clarity.
Point 3: It would be good to be consistent in the term “Vascular Endothelial Growth Factor” for VEGF, and not call it “Endothelial vascular growth factors” in Table 1.
Response 3: I apologize for this typographical error, it has now been corrected in Table1.
Point 4: Figures 2 and 3 are too big, and contain a lot of detail. Moreover, these figures are placed far from their explanations, which hampers understanding them. It would the good if the authors could address these problems.
Response 4: We thank the reviewer for this insightful comment. Indeed, Figures 2 and 3 contain a lot of detail, in order to group all the toxins mentioned in the article according to their cellular targets and their cardiovascular effects (anti-hypertensive, anti-atherosclerotic, etc.). The aim of these graphs was also to show the overlaps and interferences of the action mechanisms of the different snake venom toxins, which therefore required large summary graphs. We think that dissociating these graphs would cause the figures to lose their summary significance.
Point 5: line 248: incudes -> includes
Response 5: Thank you for pointing out these mistakes. Corrections are now included in the revised manuscript.
Moreover, the entire manuscript has also undergone extensive and substantial English editing.